# Multicomponent regulation of actin barbed end assembly by twinfilin, formin and capping protein

Heidi Ulrichs[1,2,3], Ignas Gaska [1,2,3] & Shashank Shekhar [1,2] ✉

Cells control actin assembly by regulating reactions at actin filament barbed ends. Formins accelerate elongation, capping protein (CP) arrests growth and twinfilin promotes depolymerization at barbed ends. How these distinct activities get integrated within a shared cytoplasm is unclear. Using microfluidics-assisted TIRF microscopy, we find that formin, CP and twinfilin can simultaneously bind filament barbed ends. Three-color, single-molecule experiments reveal that twinfilin cannot bind barbed ends occupied by formin unless CP is present. This trimeric complex is short-lived (~1 s), and results in dissociation of CP by twinfilin, promoting formin-based elongation. Thus, the depolymerase twinfilin acts as a pro-formin pro-polymerization factor when both CP and formin are present. While one twinfilin binding event is sufficient to displace CP from the barbed-end trimeric complex, ~31 twinfilin binding events are required to remove CP from a CP-capped barbed end. Our findings establish a paradigm where polymerases, depolymerases and cappers together tune actin assembly.

Cellular actin dynamics are essential in several key processes, such as cell migration, wound healing, cell division, and endocytosis[1,2]. Depending upon the requirements of specific processes, cells dynamically tune the rate of assembly, size, and architecture of their filamentous actin networks. For example, while formins assemble fast-elongating linear actin networks comprising of long bundled filaments, e.g., in filopodia, stereocilia, and stress fibers[3–7], the Arp2/3 complex assembles dendritic actin arrays made of short, relatively slowly growing branched actin filaments, e.g., in lamellipodia of motile cells and at sites of endocytosis[1,8,9]. How cells assemble actin networks with such diverse morphologies and dynamics in a shared cytoplasm remains an open question.

Cellular actin assembly is primarily governed by reactions occurring at the barbed end of actin filaments (sometimes also referred to as the plus end)[2,10]. Even though actin filaments can elongate by spontaneous addition of actin monomers, barbed end dynamics can be further tuned by three distinct classes of proteins that directly bind barbed ends. These include polymerases, cappers, and depolymerases. Polymerases, like formins and Ena/VASP, nucleate actin filaments, and support filament elongation by remaining processively bound to filament barbed ends[11,12]. Formins can additionally accelerate the rate of barbed-end assembly by up to fivefold in the presence of profilin-bound actin monomers (referred to as profilin-actin monomers or PA henceforth)[13,14]. Cappers, like capping protein (CP) and gelsolin, cause the complete arrest of filament growth by preventing monomer addition at free barbed ends[15,16]. Depolymerases, like twinfilin, comprise the third class of barbed-end binding proteins[17,18]. Initially discovered as actin monomer-sequestering protein[19], twinfilin induces barbed-end depolymerization in a nucleotide-specific fashion[17,18]. While twinfilin accelerates depolymerization of newly assembled (ADP-$P_i$) filaments, it slows down depolymerization of aged (ADP) actin filaments[18,20]. Notably, twinfilin's depolymerization activities persist even in cytosol-mimicking conditions, i.e., the presence of high concentrations of actin monomers[17,18,20].

Although the individual effects of formin, CP and twinfilin on actin assembly are relatively well-characterized, how they simultaneously act in multiprotein teams at barbed ends to influence actin assembly remains unclear. This question is especially important as these factors

[1]Department of Physics, Emory University, Atlanta, GA 30322, USA. [2]Department of Cell Biology, Emory University, Atlanta, GA 30322, USA. [3]These authors contributed equally: Heidi Ulrichs, Ignas Gaska. ✉e-mail: shekhar@emory.edu

are often found in the same cellular compartments, such as filopodia, lamellipodia and stereocilia[6,18,21–24]. In addition to twinfilin's monomer sequestration and barbed end depolymerization functions, it can also directly bind CP via its C-terminal tail. The CP-twinfilin interaction is required for proper intracellular localization of CP but is dispensable for twinfilin's uncapping activity in vitro[20,25]. Moreover, in absence of direct visualization of CP's uncapping by twinfilin, the underlying mechanism remains unclear. While a number of recent studies have looked at twinfilin-CP interaction, twinfilin's effects on formin have not yet been extensively studied. We previously reported that formin's rate of barbed-end elongation is not affected by a high concentration of twinfilin[18]. Nevertheless, twinfilin's effects on formin's long-lived residence time at the barbed end has yet to be investigated.

CP and formin were initially thought to bind barbed ends in a mutually exclusive fashion[26–28]. Contrary to this assumption, it was discovered that formin and CP simultaneously bind the same barbed end to form a so-called barbed-end decision complex (henceforth interchangeably referred to as BFC, where B is the barbed end, F is formin, and C is CP)[29,30]. Their concurrent presence at the filament end accelerates barbed-end dissociation of formin and CP by about 50-fold and 10-fold, respectively[29]. While the mechanisms discussed above reduce barbed-end residence times of formin and CP from 120 min and 30 min to just a few minutes[29], these time scales are still too slow to explain the rapid rates at which intracellular actin structures get assembled, arrested, and turned over in a few seconds[1,31]. This prompted us to take a fresh look at the multicomponent dynamics between CP, formin, and twinfilin at filament barbed ends.

Here we show that while twinfilin alone has no direct effects on formin's processivity, it greatly enhances formin's processivity in presence of CP (Fig. 1). Using microfluidics-assisted total internal reflection fluorescence (mf-TIRF) microscopy[32,33], we find that despite its depolymerization of free barbed ends, twinfilin effectively promotes actin assembly when both formin and CP are present together (Fig. 2). We find that formin, CP and twinfilin can all simultaneously bind a filament barbed end in a trimeric complex (Fig. 3). The dynamics of this formin-CP-twinfilin complex at the barbed end are visualized by multicolor single-molecule imaging. We discover that twinfilin reduces the lifetime of the CP-formin decision complex at the barbed end by about 17-fold. The trimeric complex formation leads to accelerated transitions between these proteins at the barbed end. We also visualize twinfilin's uncapping of CP-bound barbed ends and find that twinfilin displaces CP from the formin-CP complex much more efficiently than from barbed ends bound only to CP (Fig. 4). While only a single twinfilin binding event is sufficient to remove CP from formin-CP complexes, on average it takes about 31 twinfilin binding events to displace CP from barbed ends bound only to CP. Using separation-of-function mutants, we find that twinfilin's direct interaction with the actin filament is necessary for its ability to rescue formin's processivity (Fig. 5). This provides direct evidence of a depolymerase, polymerase, and capper simultaneously binding a growing filament end, and demonstrates the multicomponent mechanism by which they can regulate each other's barbed end activities.

## Results

### Capping protein and twinfilin differentially influence formin's processivity

We studied how CP and twinfilin influence formin's processivity at actin filament barbed ends. Fluorescent actin filaments were initiated in a mf-TIRF chamber by exposing coverslip-anchored formins (mDia1 FH1-FH2-C) to a solution containing fluorescent actin monomers and profilin-actin (PA) (Fig. 1a). Then, to confirm these filaments were indeed elongating from formins, a solution containing profilin and unlabeled actin monomers was flowed. The use of unlabeled actin

prevents any artifacts that might arise from the use of labeled actin. Since elongation occurs by insertion of unlabeled monomers at the formin-bound barbed end, the pre-existing fluorescent segment of the filament appears to move in the direction of the flow (Fig. 1a, b and Supplementary Movie 1), away from the location of formin anchoring.

Dissociation of filaments from surface-anchored formins caused the immediate disappearance of filaments from the field of view (BF → B + F, where B denotes the barbed end, F denotes the formin, and BF is the formin-bound barbed end). We recorded the disappearance of actin filaments from the field of view over time. Changes in the time-dependent survival fraction of formin-bound filaments were then used to determine the dissociation rate of formin from the barbed end. Using unlabeled actin subunits eliminates the effects of labeled actin on formin's processivity. To confirm that filament disappearance was due to detachment of filaments from formin rather than due to the detachment of the entire formin-filament complex from the glass coverslip, we re-exposed the surface to a flow containing 1 μM Alexa-488 G-actin and 0.5 μM profilin. Consistent with previous studies, over 80% of formins were able to renucleate new filaments following detachment of initially nucleated filaments[29,34].

First, we asked how CP influences the processivity of formins at filament barbed ends. Formin-elongated fluorescent filaments were exposed to a solution containing either profilin-actin alone or with a range of concentrations of CP. In control reactions, the fluorescent segment of actin filaments continued to move away from the attached formin, at a constant speed, in the direction of the flow, indicating processive elongation by formin (Fig. 1b and Supplementary Movie 1). Only a small fraction (~20%) of filaments dissociated from formins over the duration of the experiment (500 s). The average barbed-end dwell time of formin mDia1 was ~35 min in control experiments (Fig. 1f, dwell time = 1/(dissociation rate)). Formin's long barbed-end residence time measured here agrees with previous studies[13,14,34]. Interestingly, when capping protein was introduced (in the presence of PA), actin filaments rapidly dissociated from formins and disappeared from the field of view (Fig. 1c, f and Supplementary Movie 2). The rate of dissociation of formins increased with CP concentration (5 nM to 1 μM). Compared to control, 1 μM CP increased formin's rate of dissociation from the barbed end by about 30-fold (Fig. 1g).

While the cytoplasmic concentration of CP is around 1 μM[35], majority of it is thought to be sequestered by V1/myotrophin[36]. As a result, only about 10–20 nM free CP is expected to be available for binding barbed ends in cells[36]. We found that even a low concentration of CP in this range (~50 nM) was sufficient to accelerate formin dissociation by about tenfold compared to the control (Fig. 1g).

We then asked if twinfilin also influenced formin's dissociation from barbed ends. In contrast to CP, we found that the presence of up to 1 μM twinfilin (mouse mTwf1, referred to as "twinfilin" hereafter) did not significantly change formin's barbed-end dwell time (Fig. 1d, h, i). Consistent with our earlier study, twinfilin also had no observable effect on formin's rate of elongation[18] (Supplementary Fig. 1). Our results imply that unlike capping protein, formins fully protect barbed ends from twinfilin, indicating that twinfilin cannot associate with a formin-bound barbed end.

We then asked how simultaneous presence of twinfilin and CP might influence formin's processivity. When formin-nucleated filaments are exposed to CP and twinfilin at the same time (in the presence of PA), formin dissociated at a rate intermediate between that of control and capping protein (Fig. 1h, i and Supplementary Movie 3). While there was a tenfold reduction in formin's processivity in the presence of 50 nM CP, the reduction was just threefold when 50 nM CP was supplemented with 1 μM twinfilin. Our data suggest that twinfilin's presence led to a reduction in the adverse effects of CP on formin's processivity. Consistently, actin filaments also grew substantially longer prior to their detachment in the presence of twinfilin and CP together (Fig. 1e) as compared to CP alone (Fig. 1c).

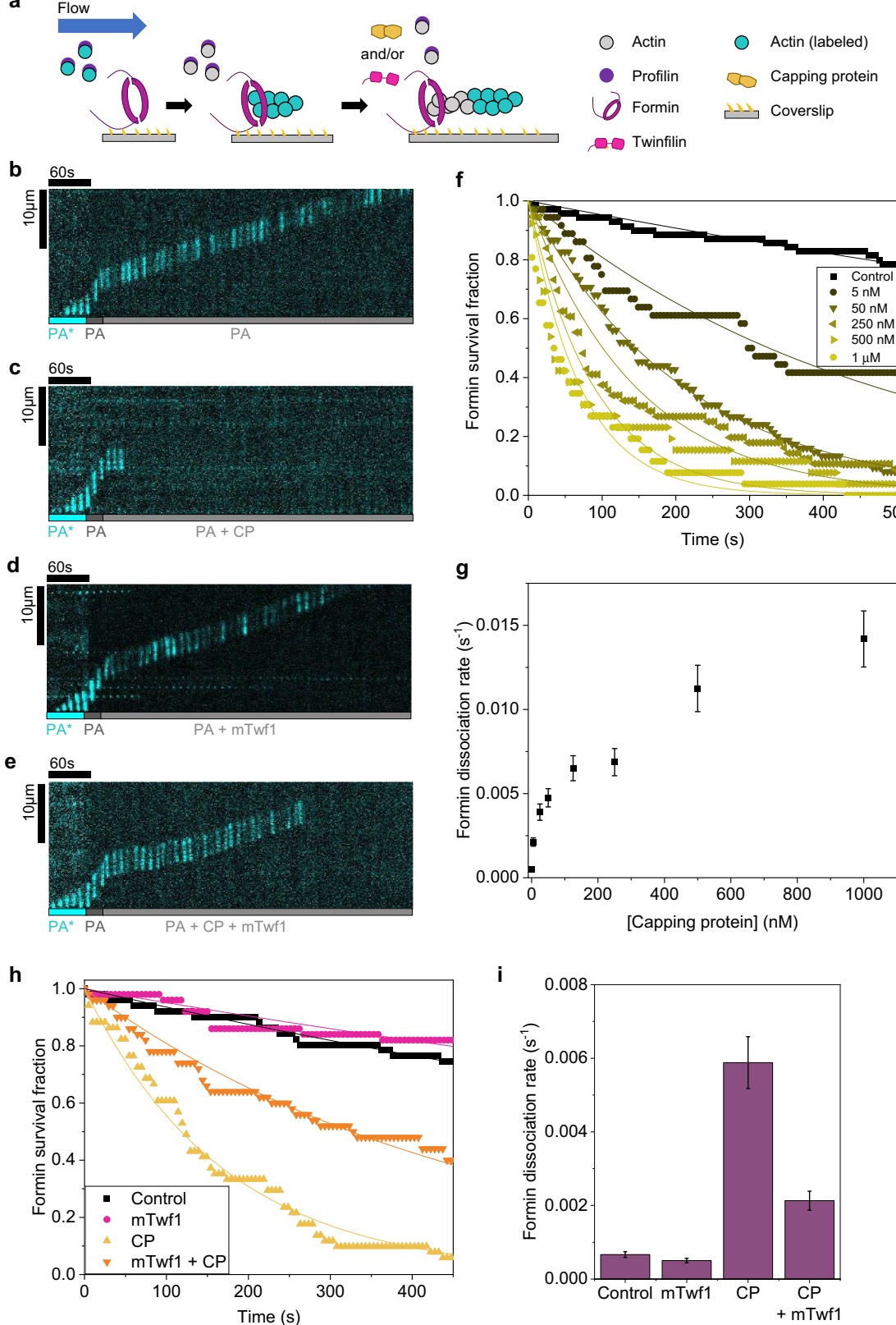

### Twinfilin accelerates dissociation of the formin–CP complex at barbed ends

X-ray diffraction and EM studies suggest that both CP and twinfilin interact directly with filament barbed ends[37–40]. Moreover, twinfilin destabilizes CP's barbed-end localization and causes a sixfold reduction in CP's barbed-end lifetime[20]. We, therefore, wondered if

twinfilin's rescue of formin's processivity from CP might be due to its destabilizing effects on CP in the formin-CP-barbed end decision complex. To investigate this, we formed decision complexes by exposing formin-nucleated fluorescent actin filaments to a high concentration of CP (1 μM). This caused an almost immediate arrest of actin filament elongation and formation of formin-CP-barbed end

**Fig. 1 | Effect of twinfilin and capping protein (CP) on processivity of formin.**
**a** Schematic of the experimental strategy. Actin filaments were nucleated from coverslip-anchored formins by a flow containing 1 μM G-actin (15% Alexa-488 labeled) and 0.5 μM profilin. Filaments were then elongated in the presence of 1 μM unlabeled G-actin and 4 μM profilin to ensure insertional elongation between fluorescent fragments and surface-anchored formins. Filaments were then exposed to a flow containing 0.2 μM unlabeled G-actin and 0.7 μM profilin (PA) (control) with or without a range of concentrations of CP and/or mTwf1 (alone or together). The survival fraction of formin-bound filaments attached was monitored as a function of time. **b** Representative kymographs of a formin-anchored filament elongating from 0.2 μM unlabeled G-actin and 0.7 μM profilin (PA) (see Supplementary Movie 1). **c** Same conditions as (**b**) but supplemented with 50 nM CP (see Supplementary Movie 2). **d** Same conditions as (**b**) but supplemented with 1 μM mTwf1. **e** Same conditions as (**b**) but supplemented with 50 nM CP and 1 μM mTwf1 (see Supplementary Movie 3). **f** Survival fraction of formin-bound filaments (BF) as a function of time in the presence of PA supplemented with a range of CP concentrations. Experimental data (symbols) are fitted to a single-exponential decay function (lines) to determine the formin dissociation rate $k_{-F}$ (BF → B + F). Number of filaments analyzed for each condition (0 to 1 μM CP): 70, 36, 75, 56, 26, 26. **g** Formin dissociation rate $k_{-F}$ as a function of CP concentration, determined from data in (**f**). **h** Survival fraction of formin-bound filaments (BF) as a function of time in the presence of PA alone (black symbols, $n = 51$ filaments) or supplemented with 1 μM mTwf1 (magenta symbols, $n = 50$ filaments), 50 nM CP (yellow symbols, $n = 51$ filaments) or 1 μM mTwf1, and 50 nM CP together (orange symbols, $n = 50$ filaments). Experimental data (symbols) are fitted to a single-exponential decay function (lines). **i** Formin dissociation rate $k_{-F}$ as determined from data in (**h**). Error bars in **g**, **i** indicate 65% confidence intervals based on fits (see Methods). Source data are provided as a Source Data file.

decision complexes[29,30]. To study the effects of twinfilin on BFC complexes, BFC complexes were then exposed to a flow containing either profilin-actin only (control) or supplemented with a range of concentrations of twinfilin (Fig. 2a).

The BFC complexes can dissociate by two different routes−(1) via dissociation of CP (BFC → BF + C, with CP departing from the BFC complex with a rate $k'_{-C}$), which leads to resumption of elongation of formin-bound filaments (Fig. 2b, top) or (2) via dissociation of formin (BFC → BC + F, with formin detaching from the BFC complex with a rate $k'_{-F}$), which leads to detachment and disappearance of the filament (Fig. 2b, bottom). We found that twinfilin dramatically accelerated disassembly of BFC complexes in a concentration-dependent manner (Fig. 2c, d). Compared to the control, 1 μM twinfilin increased the rate of BFC dissociation ($k_{-BFC}$) by about 18-fold.

While 1 μM twinfilin increased CP's dissociation rate from the BFC complex by ~11-fold, formin's rate of dissociation did not change (Fig. 2e, f). The dissociation kinetics of BFC complexes into BC and BF were exponential with observed rate constant $k_{-BFC} = k'_{-F} + k'_{-C}$ where $k'_{-F}$ is the observed rate constant of dissociation of formin from the BFC complex and $k'_{-C}$ is the observed rate constant of dissociation of CP from the BFC complex (Fig. 2e and Supplementary Fig. 2). The fraction of filaments transitioning to BF and BC states upon BFC dissociation is given by the $k'_{-C}/(k'_{-C} + k'_{-F})$ and $k'_{-F}/(k'_{-C} + k'_{-F})$ (see Methods). We derived the values $k'_{-C}$ and $k'_{-F}$ as a function of twinfilin concentration (Fig. 2e, f, and Supplementary Fig. 2). We also analyzed the route by which the BFC complexes dissociated (BFC → BF + C or BFC → BC + F). In absence of twinfilin, slightly more than half of all BFC complexes transitioned to BC (~54%) and the rest transitioned to BF (~46%). In presence of twinfilin however, this ratio was skewed heavily towards BF (~90% at 1 μM twinfilin), i.e., CP dissociated and left formin solely bound to the barbed end in majority of BFC complexes (Fig. 2g). These filaments immediately returned to rapid elongation, characteristic of formin's presence at the barbed end.

Taken together, our results indicate that although twinfilin is a depolymerase of free barbed ends, it promotes filament assembly by formin when both CP and formin are present. Just as twinfilin can uncap CP from free barbed ends[20], our results indicate twinfilin might also be able to destabilize CP from BFC complexes by forming a ternary complex, BFCT, with formin and CP at the barbed end (T in BFCT denotes twinfilin). However, due to our inability to directly visualize twinfilin, it was not possible to ascertain whether twinfilin's effects were due to its barbed-end binding or its interactions with filament sides.

## Single-molecule visualization of twinfilin's effects on BFC dynamics

To directly visualize the effects of twinfilin on the dynamics of the BFC complex, we used three-color single-molecule TIRF imaging. SNAP-tagged constructs of CP (SNAP-CP) and formin (SNAP-mDia1) were expressed and labeled with benzylguanine functionalized green-

excitable (549-CP) and red-excitable (649-mDia1) fluorescent dyes. Photobleaching data confirmed that majority of 549-CP molecules were labeled with only one dye molecule, consistent with a single SNAP-tag per heterodimeric CP molecule[30] (Supplementary Fig. 3). Photobleaching tests of 649-mDia1 showed that majority of these molecules exhibited a single- or double-step bleaching profile, consistent with the dimeric nature of formin mDia1 molecules[29,41] (Supplementary Fig. 4). Alexa-488 labeled G-actin actin monomers were incubated with 649-mDia1 in a non-microfluidic, conventional flow cell. The majority of newly nucleated filaments displayed fluorescent formins at their barbed ends (Fig. 3a). All filament barbed ends with detectable 649-mDia1 underwent rapid, continuous elongation with no noticeable pauses. Consistent with long barbed-end dwell times and low photobleaching, the majority of 649-mDia1 molecules remained bound to elongating actin filaments for the duration of the experiment.

649-mDia1 bound barbed ends were then exposed to a profilin-actin solution containing either 549-CP alone or together with unlabeled twinfilin. In the absence of twinfilin, we routinely observed the following (Fig. 3a, b): first, the 649-mDia1 was joined at the barbed end by a 549-CP molecule to form the BFC complex; second, upon arrival of 549-CP, the filament immediately stopped elongating; third, following a brief interval of time during which both molecules were jointly bound to the barbed end, the 649-mDia1:549-CP complex dissociated, and one of these two molecules departed from the barbed end, leaving the other behind (Supplementary Movie 4). In approximately 58% of the complexes, 649-mDia1 dissociated ($N = 42$ out of 72 complexes), leaving 549-CP at the barbed end, and no further elongation was observed (Fig. 3a, c). In the remaining complexes, 549-CP departed ($N = 30$ out of 72 complexes) and the filament immediately switched from the paused state to the rapidly elongating state, characteristic of formin-bound barbed ends. The 649-mDia1:549-CP barbed end complex had an average lifetime of 149 ± 12.4 s (mean ± sem) (Fig. 3e, f). In contrast, when 649-mDia1 bound filaments were exposed to 549-CP but in presence of 20 nM twinfilin, a much shorter pause was observed during which both 549-CP and 649-mDia1 were bound and the filament elongation was arrested (Fig. 3d). The average complex lifetime reduced to 35.4 ± 5 s (mean ± sem), a 75% reduction over control (Fig. 3e, f). In agreement with our mf-TIRF experiments, twinfilin also altered the outcome of BFC complex resolution. In presence of 20 nM twinfilin, about 73% of complexes (36 out of 49) transitioned to the 649-mDia1 BF state as compared to 42% of complexes (30 out of 72) in control experiments.

Our mf-TIRF and single-molecule experiments together conclusively establish that twinfilin influences both the lifetime and eventual outcome (BF or BC) of the BFC complex. We then asked if these effects are caused by binding of twinfilin along the filament length or by its interactions at the barbed end. To do this, we expressed mouse twinfilin-1 as a SNAP-tagged fusion protein and directly visualized its interactions with the BFC complex.

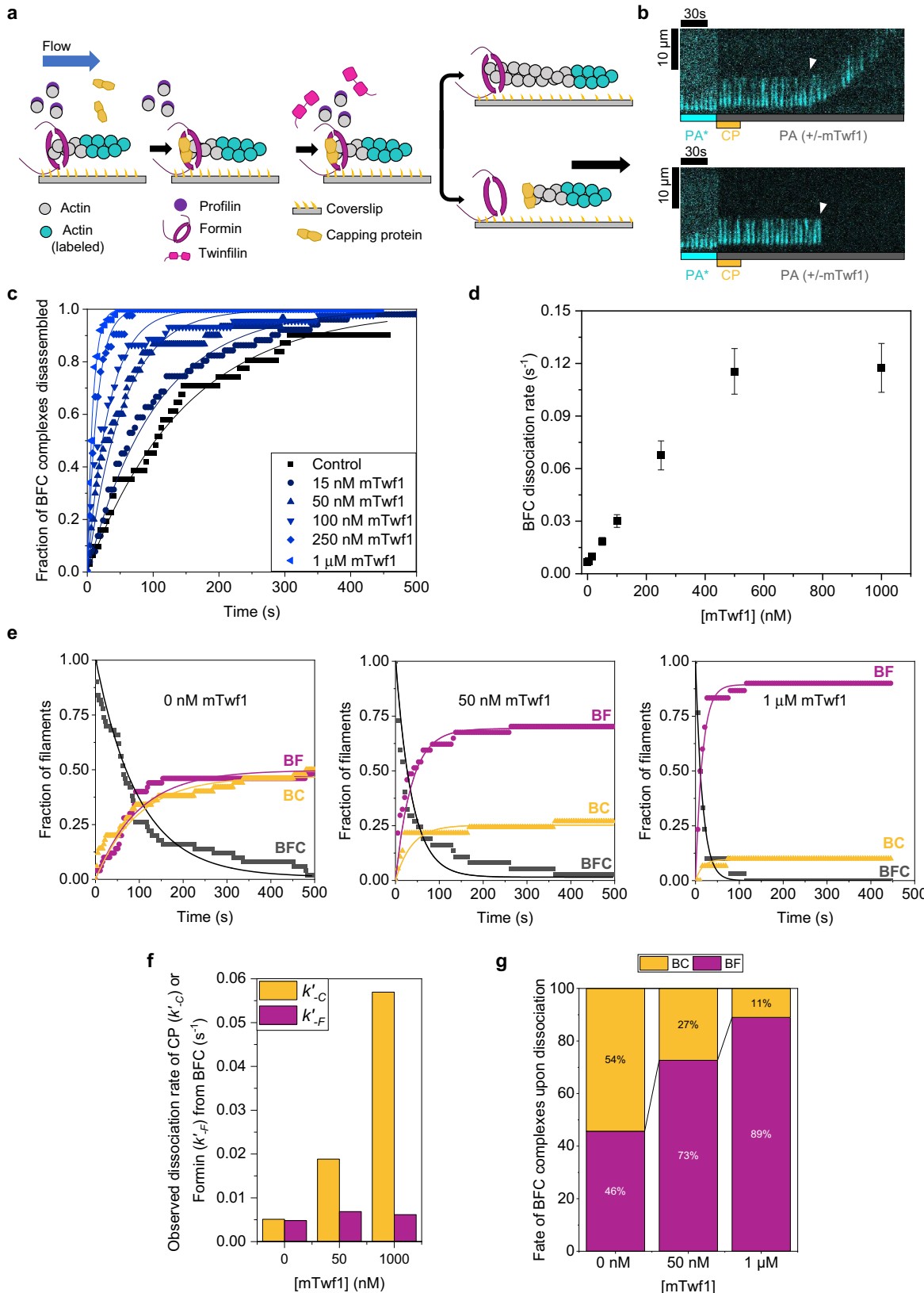

SNAP-tagging did not alter mTwf1's ability to uncap CP from CP-bound barbed ends (Supplementary Fig. 5). Photobleaching records of surface adsorbed 549-SNAP-mTwf1 showed that almost all molecules exhibited single-step photobleaching, confirming that labeled twinfilin molecules were monomeric (Supplementary Fig. 6).

Using labeled twinfilin, we further examined the mechanism by which twinfilin accelerated dissociation of BFC complexes. We exposed 649-mDia1 bound Alexa-488 actin filaments to a solution containing fluorescently labeled 549-mTwf1 and unlabeled CP. Expectedly, filaments rapidly paused due to BFC complex formation between unlabeled CP and 649-mDia1, leading to an abrupt stop in the

**Fig. 2 | Effect of twinfilin on capping protein (CP)−formin decision complex.**
**a** Experimental strategy schematic. Actin filaments were nucleated from coverslip-anchored formins by flowing 1 μM G-actin (15% Alexa-488 labeled) and 0.5 μM profilin. Filaments were then exposed to flow containing 1 μM unlabeled G-actin, 4 μM profilin, and 1 μM CP for ~10 s to convert formin-bound barbed ends (BF) to formin-CP-bound barbed ends (BF + C → BFC). BFC complexes were then exposed to flow containing PA only or with a range of mTwf1 concentrations.
**b** Representative filament kymographs are transitioning to the BFC state upon exposure to 1 μM CP. Upon CP removal from solution and exposure to PA (with or without mTwf1), filaments resume elongation following CP dissociation from BFC (top, BFC → BF + C) or detach from formin (bottom, BFC → BC + F). White arrowheads denote BFC complex dissociation. **c** Dissociation fraction of formin-CP-bound filaments (BFC) as a function of time in the presence of PA with/without a range of mTwf1 concentrations. Experimental data (symbols) are fitted to a single-exponential function (lines) to determine the rate of BFC disassembly. Number of filaments analyzed per condition (0 to 1 μM mTwf1): 31, 51, 30, 44, 42, and 50. **d** BFC dissociation rate into BC or BF as a function of mTwf1 concentration, determined from (**c**). **e** Fraction of BFC (black symbols) filaments transitioning to BF (magenta symbols) or BC (yellow symbols). Experimental data (symbols) are fitted to exponential fits (lines), such that $k_{-BFC} = k'_{-F} + k'_{-C}$, where $k'_{-F}$ is the formin dissociation rate from BFC (BFC → BC + F) and $k'_{-C}$ is CP dissociation rate from BFC (BFC → BF + C). Conditions – left (0 nM mTwf1, 50 filaments), center (50 nM mTwf1, 37 filaments), right (1 μM mTwf1, 30 filaments). See Supplementary Fig. 2 for the range of mTwf1 concentrations. **f** Dissociation rate of CP ($k'_{-C}$) or Formin ($k'_{-F}$) from BFC complexes **g** Percentages of BFC complexes in (**e**) transitioning to BC (yellow) or BF (magenta) at different mTwf1 concentrations. Error bars in (**d**) indicate 65% confidence intervals based on fits (see methods). Source data are provided as a Source Data file.

translocation of 649-mDia1 molecules bound to actin filaments. To our surprise, we observed 549-mTwf1 molecules were now able to transiently join 649-mDia1 (and unlabeled CP) at the barbed end (Fig. 3g, h). Association of 549-mTwf1 molecules with 649-mDia1:unlabeled CP-bound barbed ends were very brief, with an average dwell time of 1.4 ± 0.2 s (mean ± sd, $n = 8$ events), translating to a dissociation rate of 0.71 s$^{-1}$. The majority of 549-mTwf1:649-mDia1:unlabeled CP BFCT complexes remained intact just for a single frame implying the 0.71 s$^{-1}$ rate of dissociation of 549-mTwf1 molecules from BFCT complex is the lower bound for this rate. Importantly, the departure of 549-mTwf1 from the BFCT complex led to an immediate transition of the barbed end from the arrested state to the fast formin-based elongation state with visible translocation of the 649-mDia1 molecule. Notably, in absence of CP, we never observed colocalization of 649-mDia1 and 549-mTwf1 at barbed ends. This explains our mf-TIRF results that twinfilin does not influence the rate of elongation or processivity of formins (Fig. 1h and Supplementary Fig. 1). Together, these observations indicate that the unlabeled CP molecule and twinfilin departed the barbed end simultaneously, leaving formin behind. Notably, we never observed twinfilin and CP simultaneously arrive at formin-occupied barbed ends.

### Single-molecule analysis uncovers twinfilin's uncapping mechanism

Although twinfilin accelerates CP's dissociation[42], in the absence of simultaneous visualization of these two proteins at barbed ends, the underlying mechanism has remained obscure. We therefore transiently exposed actin filaments elongating from Alexa-488 labeled monomers to 649-CP in a non-microfluidic, conventional flow cell (Fig. 4a). Expectedly, all filaments with a visible 649-CP signal at their barbed end immediately stopped growing following 649-CP's arrival. Upon exposure to 549-mTwf1, short-lived associations of twinfilin at the 649-CP-bound barbed ends were observed. To our surprise, unlike in the case of BFC complexes where a single twinfilin binding event was sufficient to cause CP's departure, 649-CP remained bound to the barbed end despite repeated arrivals and departures of 549-mTwf1 molecules at the barbed end (Fig. 4b). Each 549-mTwf1:649-CP colocalization event lasted about a second. However, owing to the slow rate of uncapping by twinfilin (it only accelerates the uncapping rate by sixfold), capturing simultaneous departure of 649-CP and 549-mTwf1 required frequent, long-term imaging, which in turn caused excessive photobleaching of 649-CP. As a result, we observed that the disappearance of 649-CP from barbed ends was not always followed by filament depolymerization. These observations suggested that the disappearance of the 649-CP signal from barbed ends was due to photobleaching and not because of CP's dissociation.

Owing to photobleaching of 649-CP, we instead used unlabeled CP. In these experiments, we used the onset of depolymerization of barbed ends to detect the moment when 549-mTwf1 dissociated unlabeled CP from barbed ends (Fig. 4c). Successful CP dissociation required on average 30.9 ± 6.5 (mean ± sem) successive 549-mTwf1 binding events, each lasting for 1.9 ± 0.1 s (mean ± sem) (Fig. 4d, e). Notably, 549-mTwf1 intensity appeared and disappeared from the barbed end in a single step, suggesting only a single 549-mTwf1 molecule colocalized with CP at a time at the filament barbed end (Fig. 4d).

### Twinfilin's interaction with actin is essential for its effects on BFC dynamics

As mentioned earlier, twinfilin can directly bind CP as well as interact with terminal F-actin subunits of the barbed end[37,43]. Which of these interactions is responsible for twinfilin's effects on BFC complex dynamics? To answer this question, we purified two twinfilin mutants: (1) the "ADF-domain mutant", which inhibits twinfilin's interaction with actin and (2) the "tail mutant", which interferes with twinfilin's direct binding to CP[44] (Fig. 5a). Pre-formed BFC complexes (similar to the strategy used in Fig. 2a) were exposed to a solution containing either profilin-actin only or supplemented with wild-type or mutant twinfilin (Fig. 5b). Mutations in the twinfilin tail led to BFC dissociation activity partway between the control and the wildtype (Fig. 5c, d). However, the ADF-domain mutant didn't cause any visible acceleration of BFC complex dissociation, implying that direct contact between twinfilin and actin is essential for these activities. Consistently, an earlier study showed that the ADF-domain mutant is also incapable of uncapping CP from free barbed ends[20]. Taken together, our data suggests that while twinfilin's interaction with CP via its tail domain is beneficial, twinfilin's binding to actin via its ADF domains is necessary for it to be able to form the BFCT ternary complex at the barbed end and for its ability to rapidly dissociate BFC complexes.

## Discussion

Living cells can rapidly tune actin assembly in response to external stimuli[1]. The actin filament barbed end is considered to be the primary site of regulation of actin assembly. Over the last few decades, several barbed-end interacting proteins, including polymerases (e.g., formin), cappers (e.g., CP), and depolymerases (e.g., twinfilin) have been discovered. How these distinct protein activities get integrated within the cytoplasm remains an open question. On its own, CP caps free barbed ends and promotes dissociation of formin from formin-bound barbed ends[29,30]. Twinfilin promotes barbed-end depolymerization and accelerates the dissociation of CP from barbed ends[17,18,20]. Here, we have uncovered a mechanism by which twinfilin, formin, and CP simultaneously bind a filament barbed end to form a multicomponent trimeric protein complex. We also find that twinfilin does not bind actin filament ends occupied by formin mDia1 unless CP is present.

Our single-molecule experiments bring valuable insights into twinfilin's uncapping mechanism (Fig. 4). We found that unlike cofilin, which uncaps by decorating actin filament sides[45], twinfilin directly associates with CP-bound barbed ends to uncap actin filaments. Twinfilin on its own is, at best, a weak uncapper and only increases the

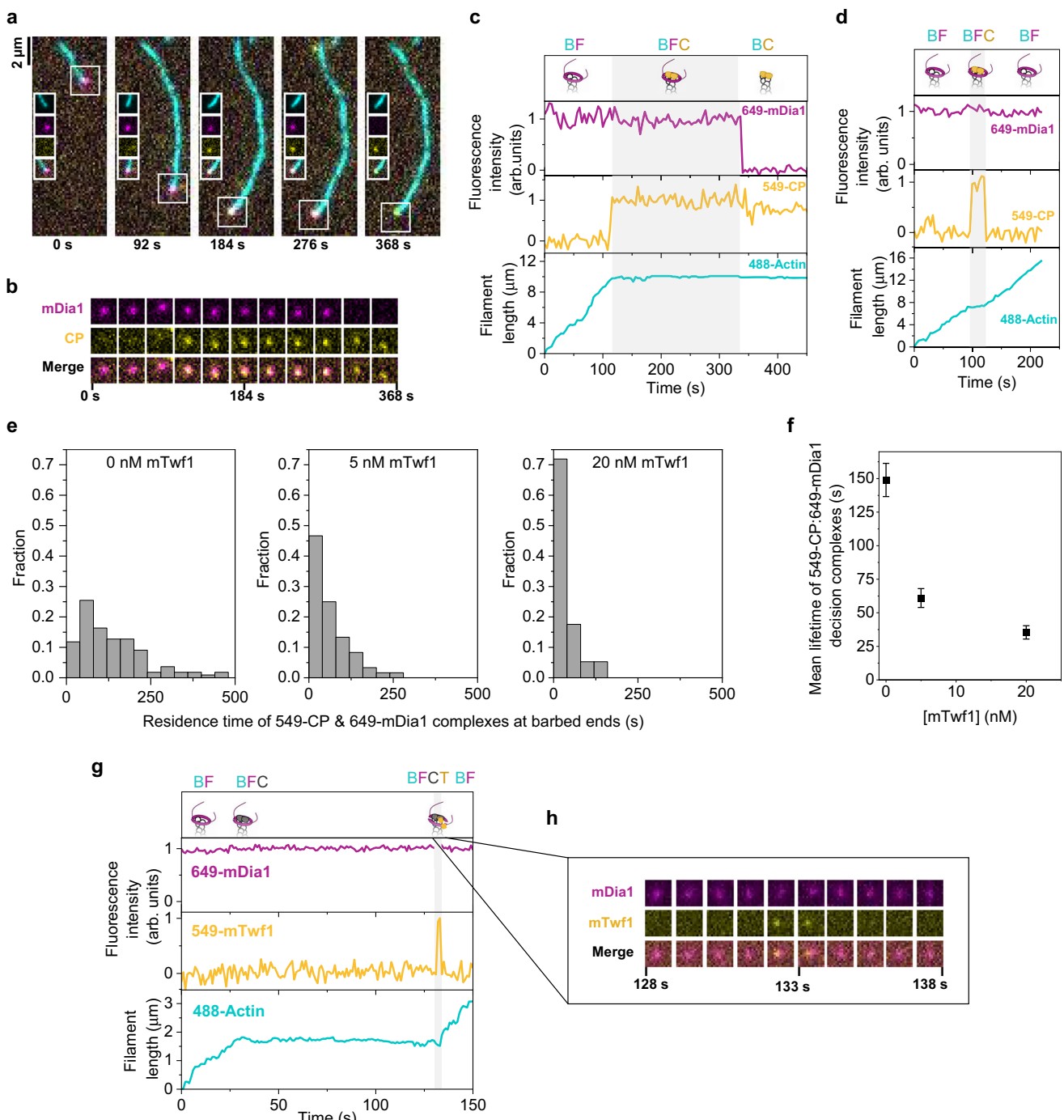

**Fig. 3 | Direct visualization of formin, CP, and mTwf1 at barbed ends.**
**a** Representative time-lapse images of a multicolor single-molecule TIRF experiment (see Supplementary Movie 4). Actin filaments were initiated by incubation of 0.5 μM G-actin (15% Alexa-488 labeled, 0.5% biotin-labeled), 1 μM profilin, and 50 pM 649-mDia1 (magenta). 649-mDia1 bound filaments were then exposed to PA and 10 nM 549-CP (yellow) with or without mTwf1. The white box around each filament end indicates the location of the barbed end. Insets show individual and merged channels localized at the barbed end. Scale bar, 2 μm. **b** Fluorescence images of a 13 × 13-pixel box around the barbed end of the filament from (**a**) show the formation and dissociation of a mDia1−CP complex at the barbed end. The Formin channel is magenta and the CP channel is yellow. **c** Fluorescence intensity and length records of the filament in (**a**) with formation and resolution of the decision complex in the absence of mTwf1. **d** Same as (**c**) but in the presence of 20 nM mTwf1. Gray shaded boxes indicate the period when both 549-CP and 649-mDia1 were simultaneously present at the barbed end, i.e., the BFC complex.

**e** Distribution of lifetimes of 649-mDia1:549-CP decision complexes at barbed ends in presence of 0 nM mTwf1 (left, *n* = 110 BFC complexes), 5 nM mTwf1 (center, *n* = 60 BFC complexes), and 20 nM mTwf1 (right, *n* = 57 BFC complexes). **f** Mean lifetimes (±sem) of 649-mDia1:549-CP decision complexes at barbed ends as a function of mTwf1 concentration, determined from data in (**e**). **g** Fluorescence intensity and length records of a filament with formation and resolution of the decision complex formed in the presence of 100 pM 649-mDia1, 10 nM unlabeled CP, and 40 nM 549-mTwf1. The gray shaded box indicates the time duration when both 549-mTwf1, 649-mDia1, and unlabeled CP were simultaneously present at the barbed end (BFCT). **h** Cropped fluorescence images of a 13 × 13-pixel box around the barbed end of the filament in (**g**) show the formation and dissociation of a 649-mDia1−unlabeled CP complex in the presence of 549-mTwf1. The formin channel is magenta and twinfilin channel is yellow. Source data are provided as a Source Data file.

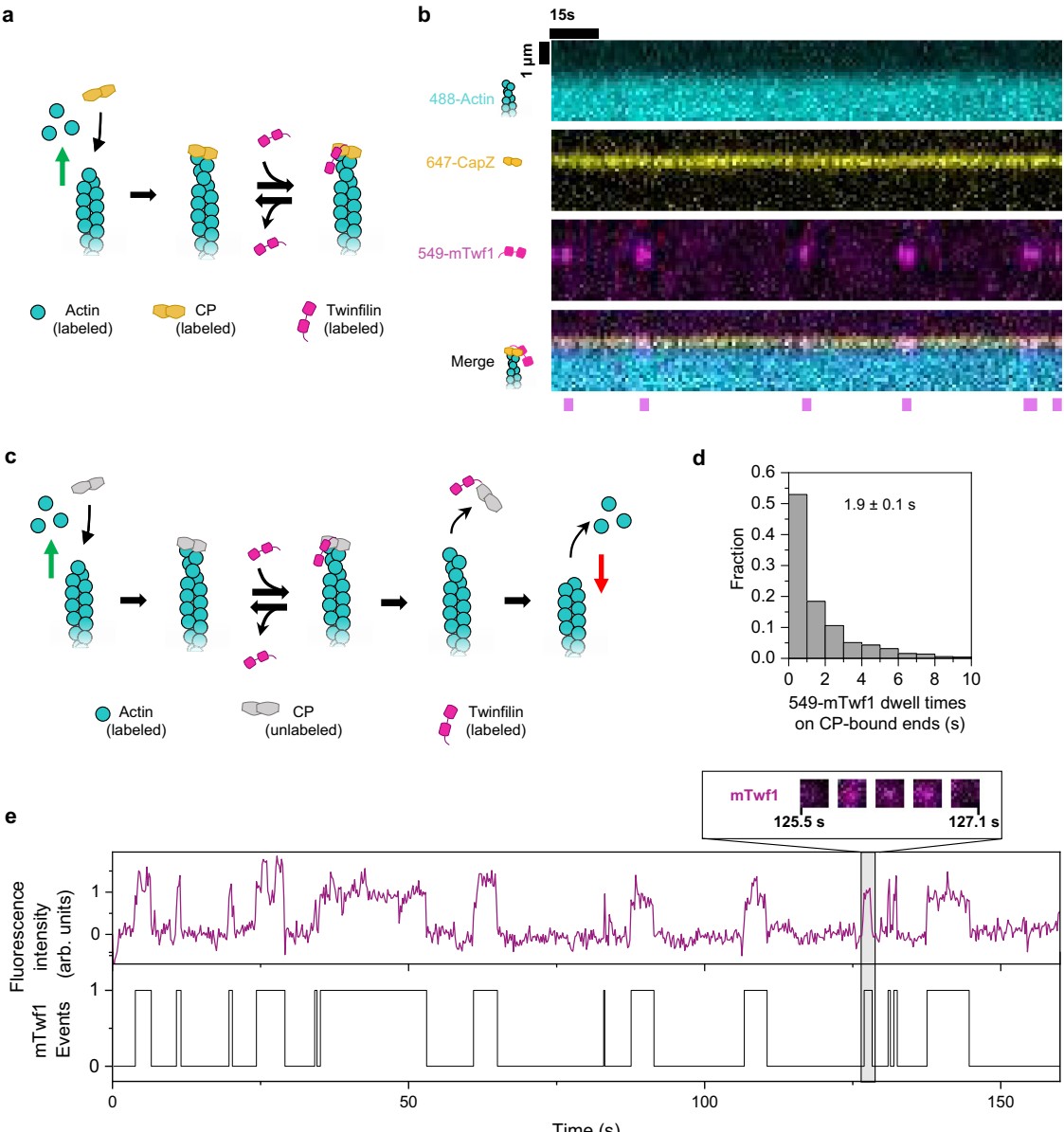

**Fig. 4 | Visualization and characterization of twinfilin's interactions with CP-bound barbed ends. a** Schematic of three-color single-molecule experiments with labeled CP and labeled mTwf1. Actin filaments were assembled from of 1 μM G-actin (15% Alexa-488 labeled, 1.4% biotin-labeled) and then capped by 10 nM 649-CP. These filaments were then exposed to 20 nM 549-mTwf1. The green arrow denotes polymerization. **b** Binding of labeled twinfilin on CP-bound filament barbed ends recorded at 1 s time resolution. Kymographs show Alexa-488 actin (top), 649-CP (second from top), 549-mTwf1 (third from top), and merge (bottom). Magenta bars denote episodes in which a 549-mTwf1 molecule was present at the 649-CP-bound barbed end of the filament. **c** Schematic of the two-color single-molecule experiments with unlabeled CP and labeled mTwf1. Actin filaments were polymerized from 1 μM G-actin (15% Alexa-488 labeled, 1.4% biotin-labeled) and 0.5 μM profilin, then capped by unlabeled CP. The capped filaments were then exposed to 15 nM 549-mTwf1. Arrival and departure of 549-mTwf1 molecules at the barbed end were

recorded until the filament started depolymerizing, i.e., CP dissociated. The green arrow denotes polymerization, and the red arrow denotes depolymerization. **d** Distribution of residence times of 549-mTwf1 on unlabeled CP-bound filament barbed ends (*n* = 510 binding events across 16 filaments). Mean dwell time = 1.9 ± 0.1 s (±sem). The histogram represents 99% of all binding events (remaining 1% outliers are not shown). 100% of binding events were included in calculations of mean dwell time. **e** Top: time records of 549-mTwf1 fluorescence intensity at the unlabeled CP-bound barbed end of an actin filament. Intensity is integrated over a 5 × 5-pixel square centered around the barbed end of the filament. Bottom: presence (1) or absence (0) of 549-mTwf1 molecules at the CP-649-bound filament barbed end shown above. Inset: cropped fluorescence images of a 10 × 10-pixel box around the barbed end of the filament showing the arrival, presence, and departure of a single 549-mTwf1 molecule at the CP-bound barbed end. Source data are provided as a Source Data file.

rate of uncapping by about sixfold. A successful uncapping event, on average, takes about 31 separate twinfilin association and disassociation events at the barbed end. In contrast, CARMIL accelerates CP's departure by about 180-fold, bringing down CP's barbed-end dwell time from 30 min to just about 10 s[46]. As a result, although both CARMIL and twinfilin bind CP similarly via their CPI motifs[47], they exert vastly different control on CP.

In addition to uncapping, twinfilin is also an actin depolymerase[17,18]. How does it carry out these two distinct activities? Our experiments with twinfilin variants containing mutations in the actin-binding ADF domains suggest that twinfilin's interactions with the actin filament are essential for it to rescue formin's processivity from CP. Mutations in twinfilin's tail domain interfere with direct twinfilin-CP[20] binding. Although less strongly than the wildtype, we

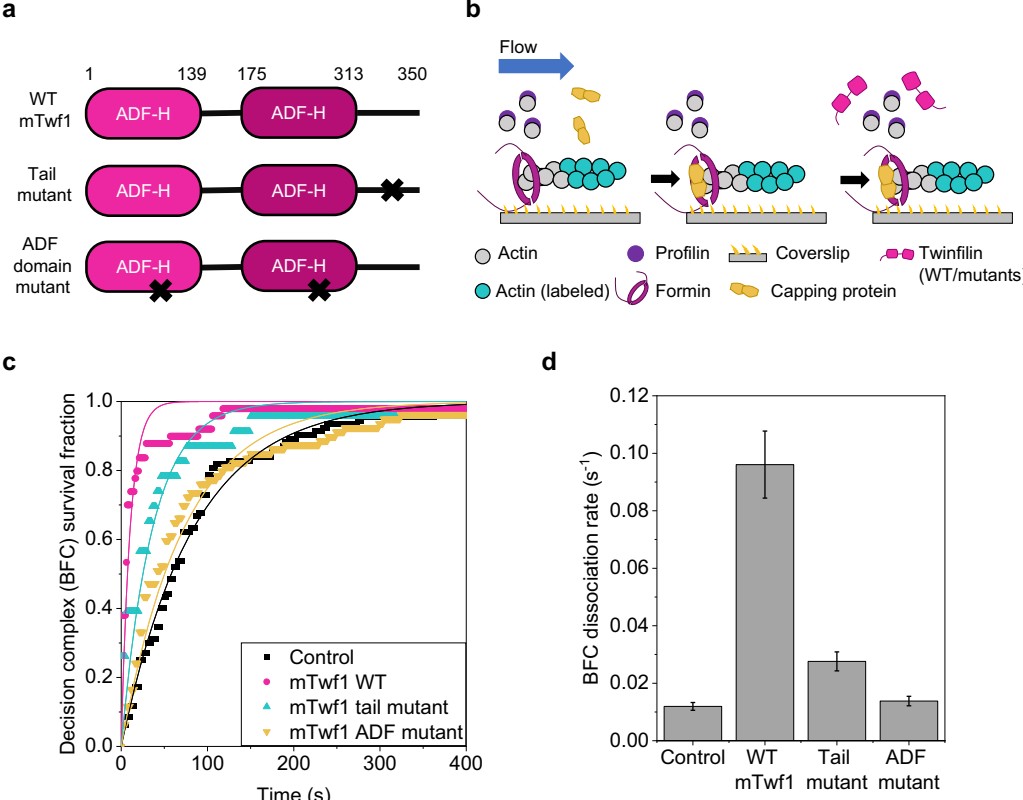

**Fig. 5 | Twinfilin's direct interaction with actin filament is essential for its effects on formin-CP decision complexes. a** Domain diagram of wild-type and mutant mTwf1 constructs used here[20]. "x" denotes the location of mutations. **b** Schematic representation of the experimental strategy. Actin filaments were nucleated from coverslip-anchored formins by introducing a flow containing 1 μM G-actin (15% Alexa-488 labeled) and 0.5 μM profilin. The filaments were then exposed to a flow containing 1 μM unlabeled G-actin, 4 μM profilin, and 500 nM CP for about 10 s to convert formin-bound barbed ends (BF) to formin-CP-bound barbed ends or decision complexes (BF + C → BFC). These BFC complexes were then exposed to a flow containing PA only or supplemented with wildtype or mutant mTwf1. **c** Survival fraction of formin-CP-bound filaments (BFC complexes) as a function of time in the presence of PA only (black symbols, 92 filaments), or supplemented with 1 μM wild-type mTwf1 (black, 36 filaments), mTwf1 ADF mutant (green, 77 filaments), or mTwf1 tail mutant (blue symbols, 22 filaments). Experimental data (symbols) were fitted to a single-exponential function (lines) to determine BFC dissociation rate $k_{-BFC}$. Error bars indicate 65% confidence intervals based on fits (see methods). **d** BFC dissociation rate for wild-type and mutant mTwf1, determined from data in (**c**). Error bars in (**d**) indicate 65% confidence intervals based on fits (see Methods). Source data are provided as a Source Data file.

find that the tail mutant can still dissociate CP from the formin-CP barbed end complex and modestly rescue formin's processivity from CP. This can be interpreted in two ways. Firstly, for twinfilin's effects on the BFC complex, its direct binding to CP is beneficial but not necessary. Interestingly, a previous study found that the tail mutant uncaps CP from free barbed ends much faster than the wildtype[20]. Alternatively, this could also mean that there are additional sites in twinfilin, apart from its tail, which participate in its direct binding to CP. Indeed, the linker region between the two ADF domains in twinfilin has been proposed as an extra site of contact between CP and twinfilin when both are bound to actin[37]. Importantly, mutations in ADF domains, which interfere with twinfilin's actin binding and extinguish twinfilin's uncapping activities at free barbed ends[20], also extinguish twinfilin's effects on BFC complex dissociation. Together, these results indicate that in the ternary BFCT complex comprising of formin, CP, and twinfilin at barbed ends, each factor is directly interacting with the actin filament and not via each other.

How would this multicomponent mechanism be relevant in vivo? Twinfilin can depolymerize free barbed ends even in the presence of polymerizable actin monomers[17,18,20]. In such cytosol-mimicking conditions, the simultaneous presence of twinfilin and CP would strongly favor filament capping and actin disassembly over actin assembly. How, then, would intracellular actin assembly occur at all? Combining our observations with previous studies, we present a working model for how polymerases, depolymerases, and cappers can simultaneously

regulate actin assembly (Fig. 6). Eventual barbed-end outcomes would depend upon whether the barbed end was bound to a formin or not. Free barbed ends would first rapidly get capped by CP (Fig. 6a). Following uncapping by twinfilin, barbed ends would undergo twinfilin-mediated depolymerization[20]. Formin-bound barbed ends, on the other hand, would first get paused by CP, leading to the formation of BFC decision complexes (Fig. 6b)[29,30]. Twinfilin's subsequent binding to BFC complexes to form BFCT complexes would cause CP's dissociation and resumption of formin-based filament elongation. As the filament ages, other actin-binding proteins with a preference for ADP-F-actin, e.g., cofilin and cyclase-associated protein (CAP), would initiate filament severing and pointed-end depolymerization[45,48–50]. As a result, filaments with free barbed ends would rapidly depolymerize from both ends into monomers and formin-bound filaments would continue polymerizing at their barbed ends while depolymerizing from their pointed ends, akin to treadmilling. Therefore, the depolymerase twinfilin can promote polymerization in the presence of both CP and formin. Although surprising, twinfilin's ability to promote assembly is not unique. Twinfilin is part of the ADF homology (ADF-H) family of proteins and contains two ADF domains[51]. Like twinfilin, cofilin also promotes uncapping and accelerates filament depolymerization (on top of its severing activities)[45,50], which in turn promotes assembly by replenishing the pool of monomeric actin.

Where in a cell would the mechanisms uncovered here be relevant? Twinfilin, CP, and formin operate simultaneously in a number of

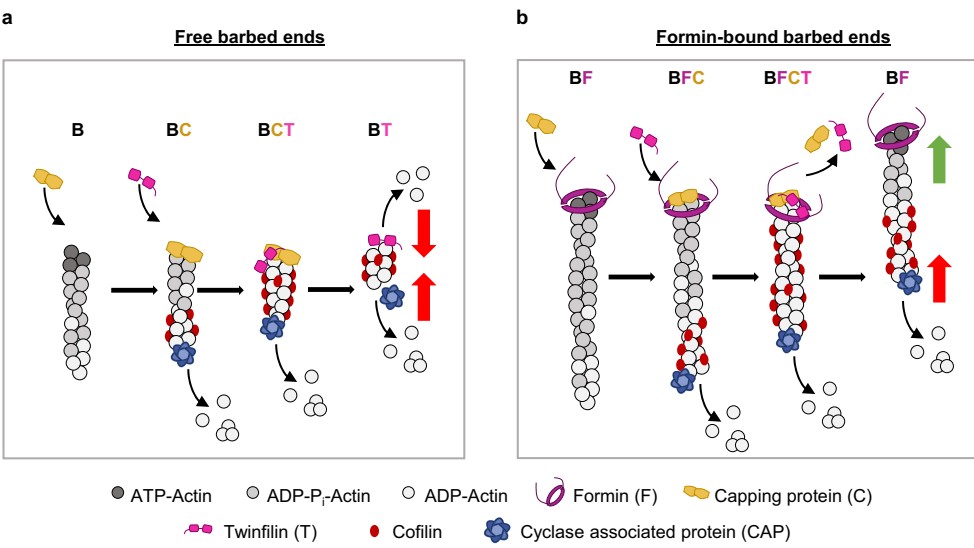

**Fig. 6 | Working model for regulation of actin dynamics by twinfilin, formin, and CP.** Barbed-end outcomes would depend upon whether barbed ends were free or formin-bound. **a** Free barbed ends (B) would rapidly get capped by CP (C), followed by CP's dissociation by twinfilin (T). This would leave twinfilin alone at the filament end, causing its depolymerization. At the same time, cofilin would bind the sides of the aging filament and synergize with cyclase-associated protein (CAP) to initiate the filament's pointed-end depolymerization. The simultaneous depolymerization at the two ends would result in the complete disassembly of the filament into monomers. **b** Formin (F) bound barbed ends would get paused by CP to form BFC complexes. Twinfilin's binding to BFC complexes would cause CP's dissociation and renewal of formin-based filament elongation. As a result, the filament would appear to a treadmill, i.e., continue elongating at the barbed end while at the same time being disassembled at the pointed end by CAP-cofilin synergy. "B" denotes barbed end, "F" denotes formin, "C" denotes CP, and "T" denotes twinfilin.

cellular compartments, including filopodia, lamellipodia, and stereocilia[6,18,21–24] where size and elongation rates of actin filaments are tightly controlled. Given the high barbed-end affinity and intracellular concentration of barbed-end binding proteins like formin, CP, twinfilin, and Ena/VASP, we expect that newly nucleated free barbed ends initiated by Arp2/3 complex would be rapidly captured by one of these factors. Although formins are mainly thought of in the context of linear actin structures, formins like FMNL2 have been shown to play a key role in branched actin networks at the leading edge[24]. In light of our results, we speculate that majority of growing barbed ends in cells might be protected by formins or Ena/VASP. In the future, it will be interesting to explore if other biochemical or mechanical factors which influence activities of CP and/or formin, e.g., CARMIL[16,36], V1/myotrophin[36], Spire[52], Bud14[53], IQGAP[54,55], or Hof1[56], polyphosphoinositides[42,57] or force[34] might also influence BFC complex dynamics to tune actin assembly. It will also be important to examine if the interactions between twinfilin, CP, and formin reported here might also play a role in human diseases and disorders, including cancer invasion and progression[58,59], hearing loss[60] as well as neuropathies and cardiac conditions[61] in which these proteins have been implicated.

## Methods

### Purification and labeling of actin
Rabbit skeletal muscle actin was purified from acetone powder generated from frozen ground hind leg muscle tissue of young rabbits (PelFreez, USA). Lyophilized acetone powder stored at −80 °C was mechanically sheared in a coffee grinder, resuspended in G-buffer (5 mM Tris-HCl pH 7.5, 0.5 mM Dithiothreitol (DTT), 0.2 mM ATP, and 0.1 mM CaCl$_2$), and cleared by centrifugation for 20 min at 50,000 × $g$. The supernatant was collected and further filtered with the Whatman paper. Actin was then polymerized overnight at 4 °C, slowly stirring, by the addition of 2 mM MgCl$_2$ and 50 mM NaCl to the filtrate. The next morning, NaCl powder was added to a final concentration of 0.6 M and stirring was continued for another 30 min at 4 °C. Then, F-actin was pelleted by centrifugation for 150 min at 280,000 × $g$, the pellet was solubilized by Dounce homogenization and dialyzed against G-buffer

for 48 h at 4 °C. Monomeric actin was then precleared at 435,000 × $g$ and loaded onto a Sephacryl S-200 16/60 gel-filtration column (Cytiva, USA) equilibrated in G-Buffer. Fractions containing actin were stored at 4 °C.

To biotinylate actin, purified G-actin was first dialyzed overnight at 4 °C against G-buffer lacking DTT. The monomeric actin was then polymerized by the addition of an equal volume of 2X labeling buffer (50 mM imidazole pH 7.5, 200 mM KCl, 0.3 mM ATP, 4 mM MgCl$_2$). After 5 min, the actin was mixed with a fivefold molar excess of NHS-XX-Biotin (Merck KGaA, Germany) and incubated in the dark for 15 h at 4 °C. The F-actin was pelleted as above, and the pellet was rinsed with G-buffer, then homogenized with a Dounce and dialyzed against G-buffer for 48 h at 4 °C. Biotinylated monomeric actin was purified further on a Sephacryl S-200 16/60 gel-filtration column as above. Aliquots of biotin-actin were snap frozen in liquid N$_2$ and stored at −80 °C.

To fluorescently label actin, G-actin was polymerized by dialyzing overnight against a modified F-buffer (20 mM PIPES pH 6.9, 0.2 mM CaCl$_2$, 0.2 mM ATP, and 100 mM KCl)[32]. F-actin was incubated for 2 h at room temperature with a fivefold molar excess of Alexa-488 NHS ester dye (Thermo Fisher Scientific, USA). F-actin was then pelleted by centrifugation at 450,000 × $g$ for 40 min at room temperature, and the pellet was resuspended in G-buffer, homogenized with a Dounce, and incubated on ice for 2 h to depolymerize the filaments. The monomeric actin was then re-polymerized on ice for 1 h by the addition of 100 mM KCl and 1 mM MgCl$_2$. F-actin was once again pelleted by centrifugation for 40 min at 450,000 × $g$ at 4 °C. The pellet was homogenized with a Dounce and dialyzed overnight at 4 °C against 1 L of G-buffer. The solution was precleared by centrifugation at 450,000 × $g$ for 40 min at 4 °C. The supernatant was collected, and the concentration and labeling efficiency of actin was determined.

### Purification and labeling of mTwf1 polypeptides
Wildtype and mutant mouse mTwf1 plasmids were a gift from Pekka Lappalainen[20]. All of these proteins were expressed in *E. coli* BL21 (pRare). Cells were grown in Terrific Broth to log phase at 37 °C.

Expression was induced overnight at 18 °C by the addition of 1 mM IPTG. Cells were harvested by centrifugation at 11,200 × g for 15 min and the cell pellets were stored at −80 °C. For purification, frozen pellets were thawed and resuspended in 35 mL lysis buffer (50 mM sodium phosphate buffer pH 8, 20 mM imidazole, 300 mM NaCl, 1 mM DTT, 1 mM PMSF, and protease inhibitors (pepstatin A, antipain, leupeptin, aprotinin, and chymostatin, 0.5 μM each)). Cells were lysed using a tip sonicator while kept on ice. The cell lysate was then centrifuged at 120,000 × g for 45 min at 4 °C. The supernatant was then incubated with 1 mL of Ni-NTA beads (Qiagen, USA) while rotating for 2 h at 4 °C. The beads were then washed three times with the wash buffer (50 mM sodium phosphate buffer pH 8, 300 mM NaCl, 20 mM imidazole, and 1 mM DTT). The beads were then transferred to a disposable column (Bio-Rad, USA). Protein was eluted using the elution buffer (50 mM phosphate buffer pH 8, 300 mM NaCl, 250 mM imidazole, and 1 mM DTT). Fractions containing the protein were concentrated and loaded onto a size exclusion Superdex 75 Increase 10/300 column (Cytiva, USA) pre-equilibrated with 20 mM HEPES pH 7.5, 1 mM EDTA, 50 mM KCl, and 1 mM DTT. Peak fractions were collected, concentrated, aliquoted, and flash-frozen in liquid $N_2$ and stored at −80 °C.

Mouse His-SNAP-mTwf1 plasmid was ordered from Twist Biosciences. SNAP-mTwf1 was purified using the same protocol as above. Purified SNAP-mTwf1 was incubated with 5x excess of SNAP-surface-549 dye (New England Biolabs, Ipswich, MA) overnight at 4 °C. Free dye was removed using a PD-10 desalting column (Cytiva, USA). Labeled protein was collected, concentrated, aliquoted, and flash-frozen in liquid $N_2$ and stored at −80 °C.

### Purification, labeling, and biotinylation of formin mDia1

Mouse his-tagged mDia1 (FH1-FH2-C) formin was expressed in *E. coli*; BL21(DE3) pLysS cells. Cells were grown in Terrific Broth to log phase at 37 °C. Expression was induced overnight at 18 °C by the addition of 1 mM IPTG. Cells were harvested by centrifugation at 11,200 × g for 15 min and the cell pellets were stored at −80 °C. For purification, frozen pellets were thawed and resuspended in 35 mL lysis buffer (50 mM sodium phosphate buffer pH 8, 20 mM imidazole, 300 mM NaCl, 1 mM DTT, 1 mM PMSF, and protease inhibitors (0.5 μM each of pepstatin A, antipain, leupeptin, aprotinin, and chymostatin)). Cells were lysed using a tip sonicator while being kept on ice. The cell lysate was then centrifuged at 120,000 × g for 45 min at 4 °C. The supernatant was then incubated with 1 mL of Ni-NTA beads (Qiagen, USA) while rotating for 2 h at 4 °C. The beads were then washed three times with the wash buffer (50 mM sodium phosphate buffer pH 8, 300 mM NaCl, 20 mM imidazole, and 1 mM DTT) and were then transferred to a disposable column (Bio-Rad, USA). Protein was eluted using the elution buffer (50 mM phosphate buffer pH 8, 300 mM NaCl, 250 mM imidazole, and 1 mM DTT). Fractions containing the protein were concentrated and loaded onto a size exclusion Superdex 200 increase 10/300 GL column (Cytiva, USA) pre-equilibrated with 20 mM HEPES pH 7.5, 150 mM KCl, 10% glycerol, and 0.5 mM DTT. Peak fractions were collected, aliquoted, and flash-frozen in liquid $N_2$ and stored at −80 °C.

SNAP-mDia1[30] was expressed and purified using the protocol above. Purified SNAP-mDia1 was incubated with 5x excess of SNAP-surface-649 dye (New England Biolabs, USA) overnight at 4 °C. Free dye was removed using a Superdex 200 increase 10/300 GL column (Cytiva, USA). Labeled protein was collected, concentrated, aliquoted, and flash-frozen in liquid $N_2$ and stored at −80 °C.

Biotin-SNAP-mDia1 was prepared by incubating purified SNAP-mDia1 with Benzylguanine-Biotin (New England Biolabs, USA) according to the manufacturer's instructions. Free biotin was removed using size-exclusion chromatography by loading the labeled protein on a Superose 6 gel-filtration column (GE Healthcare, Pittsburgh, PA) eluted with 20 mM HEPES pH 7.5, 150 mM KCl, and 0.5 mM DTT.

### Purification and labeling of capping protein

Mouse his-tagged capping protein was expressed in *E. coli*; BL21(DE3) pLysS cells. Capping protein subunits α1 and β2 were expressed from the same plasmid with a single His-tag on the alpha subunit[29]. Cells were grown in Terrific Broth to log phase at 37 °C. Expression was induced overnight at 18 °C by the addition of 1 mM IPTG. Cells were harvested by centrifugation at 11,200 × g for 15 min and the cell pellets were stored at −80 °C. For purification, frozen pellets were thawed and resuspended in 35 mL lysis buffer (50 mM sodium phosphate buffer pH 8, 20 mM imidazole, 300 mM NaCl, 1 mM DTT, 1 mM PMSF, and protease inhibitors (0.5 μM each of pepstatin A, antipain, leupeptin, aprotinin, and chymostatin)). Cells were lysed using a tip sonicator while being kept on ice. The cell lysate was then centrifuged at 120,000 × g for 45 min at 4 °C. The supernatant was incubated with 1 mL of Ni-NTA beads (Qiagen, USA) while rotating for 2 h at 4 °C. The beads were then washed three times with the wash buffer (50 mM sodium phosphate buffer pH 8, 300 mM NaCl, 20 mM imidazole, and 1 mM DTT) and transferred to a disposable column (Bio-Rad, USA). Protein was eluted using elution buffer (50 mM phosphate buffer pH 8, 300 mM NaCl, 250 mM Imidazole, and 1 mM DTT). Fractions containing the protein were concentrated and loaded onto a size exclusion Superdex 75 Increase 10/300 column (Cytiva, USA) pre-equilibrated with 20 mM Tris-HCl, 50 mM KCl, and 1 mM DTT. Peak fractions were collected, concentrated, aliquoted, and flash-frozen in liquid $N_2$ and stored at −80 °C.

SNAP-CP was expressed from a single plasmid containing His- and SNAP-tagged β1 subunit and untagged α1 subunit[30]. It was purified using the protocol above. Purified SNAP-CP was incubated with 5x excess of SNAP-surface-549 dye or SNAP-surface-649 dye (New England Biolabs, USA) overnight at 4 °C. Free dyes were removed using PD-10 desalting columns (Cytiva, USA). Labeled protein was collected, concentrated, aliquoted, and flash-frozen in liquid $N_2$ and stored at −80 °C.

### Purification of profilin

Human profilin-1 was expressed in *E. coli* strain BL21 (pRare) to log phase in LB broth at 37 °C and induced with 1 mM IPTG for 3 h at 37 °C. Cells were then harvested by centrifugation at 15,000 × g at 4 °C and stored at −80 °C. For purification, pellets were thawed and resuspended in 30 mL lysis buffer (50 mM Tris-HCl pH 8, 1 mM DTT, 1 mM PMSF protease inhibitors (0.5 μM each of pepstatin A, antipain, leupeptin, aprotinin, and chymostatin)) was added, and the solution was sonicated on ice by a tip sonicator. The lysate was centrifuged for 45 min at 120,000 × g at 4 °C. The supernatant was then passed over 20 ml of Poly-L-proline conjugated beads in a disposable column (Bio-Rad, USA). The beads were first washed at room temperature in wash buffer (10 mM Tris pH 8, 150 mM NaCl, 1 mM EDTA, and 1 mM DTT) and then washed again with two column volumes of 10 mM Tris pH 8, 150 mM NaCl, 1 mM EDTA, 1 mM DTT, and 3 M urea. Protein was then eluted with five column volumes of 10 mM Tris pH 8, 150 mM NaCl, 1 mM EDTA, 1 mM DTT, and 8 M urea. Pooled and concentrated fractions were then dialyzed in 4 L of 2 mM Tris pH 8, 0.2 mM EGTA, 1 mM DTT, and 0.01% NaN$_3$ (dialysis buffer) for 4 h at 4 °C. The dialysis buffer was replaced with fresh 4 L buffer and the dialysis was continued overnight at 4 °C. The protein was centrifuged for 45 min at 450,000 × g at 4 °C, concentrated, aliquoted, flash-frozen in liquid $N_2$, and stored at −80 °C.

### Conventional TIRF microscopy for single-molecule imaging

Glass coverslips (60 × 24 mm; Thermo Fisher Scientific, USA) were first cleaned by sonication in detergent for 20 min, followed by successive sonications in 1 M KOH, 1 M HCl, and ethanol for 20 min each. Coverslips were then washed extensively with H$_2$O and dried in an N$_2$ stream. The cleaned coverslips were coated with 2 mg/mL methoxy-polyethylene glycol (mPEG)-silane MW 2000 and 2 μg/mL biotin-PEG-

silane MW 3400 (Laysan Bio, USA) in 80% ethanol (pH 2.0) and incubated overnight at 70 °C. Flow cells were assembled by rinsing PEG-coated coverslips with water, drying with $N_2$, and adhering to µ-Slide VI0.1 (0.1 mm × 17 mm × 1 mm) flow chambers (Ibidi, Germany) with double-sided tape (2.5 cm × 2 mm × 120 µm) and epoxy resin for 5 min (Devcon, USA). Before each reaction, the flow cell was sequentially incubated for 1 min each with 4 µg/ml streptavidin and 1% BSA in 20 mM HEPES pH 7.5, and 50 mM KCl. The flow cell was then equilibrated with TIRF buffer (10 mM imidazole, pH 7.4, 50 mM KCl, 1 mM $MgCl_2$, 1 mM EGTA, 0.2 mM ATP, 10 mM DTT, 2 mM DABCO, and 0.5% methylcellulose [4000 cP]).

For 649-mDia1, 549-CP, and 488-actin experiments (Fig. 3a–f), 0.5 µM G-actin (15% Alexa-488 labeled, 0.5% biotin-labeled), 1 µM profilin, along with 50 pM 649-mDia1 in TIRF buffer were introduced into the flow cell and filaments were allowed to grow for 2 to 3 min. The flow cell was then rinsed with TIRF buffer to remove free formins, and the solution was replaced with profilin-actin and 10 nM 549-CP (with or without mTwf1). In experiments with fluorescently labeled twinfilin (Fig. 3g, h), 100 pM mDia1 was introduced to the flow cell, followed by flows containing 10 nM unlabeled CP along with 40 nM 549-mTw1. Time-lapse images were acquired every 4 s.

For three-color uncapping experiments with 549-mTwf1, 649-CapZ, and 488-actin (Fig. 4a, b), actin filaments were elongated from 1 µM G-actin (15% Alexa-488 labeled and 1.4% biotinylated G-actin). Free actin monomers were removed by rinsing the flow cell with an excess of TIRF buffer and the filaments were exposed to a solution containing 10 nM 649-CP. Following capping, the chamber was once again rinsed with TIRF buffer to remove free 649-CP and then 20 nM 549-mTwf1 was introduced. Time-lapse images were acquired every 1 s.

For two-color experiments involving 549-mTwf1, unlabeled CapZ, and 488-actin (Fig. 4c–e), actin filaments were elongated from 1 µM G-actin (15% Alexa-488 labeled and 1.4% biotinylated G-actin) and 0.5 µM profilin. Free profilin and actin monomers were removed by rinsing the flow cell with an excess of TIRF buffer and the filaments were exposed to a solution containing 10 nM unlabeled CP and 1 µM G-actin (15% Alexa-488 labeled and 1.4% biotinylated G-actin) with 0.5 µM profilin. Following capping, the chamber was once again rinsed with TIRF buffer to remove free unlabeled CP, actin, profiling, and then 15 nM 549-mTwf1 was introduced. Time-lapse images were acquired either every 200 ms or every 300 ms.

## Microfluidics-assisted TIRF (mf-TIRF) imaging and analysis

Actin filaments were first assembled in microfluidics-assisted TIRF (mf-TIRF) flow cells[32,33]. Coverslips were first cleaned by sonication in Micro90 detergent for 20 min, followed by successive 20 min sonications in 1 M KOH, 1 M HCl, and 200-proof ethanol for 20 min each. Washed coverslips were then stored in fresh 200-proof ethanol. Coverslips were then washed extensively with $H_2O$ and dried in an $N_2$ stream. These dried coverslips were coated with 2 mg/mL methoxy-poly(ethylene glycol) (mPEG)-silane MW 2000 and 2 µg/mL biotin-PEG-silane MW 3400 (Laysan Bio, USA) in 80% ethanol (pH 2.0) and incubated overnight at 70 °C. A 40 µm high PDMS mold with three inlets and one outlet was mechanically clamped onto a PEG-Silane-coated coverslip. The chamber was then connected to a Maesflo microfluidic flow-control system (Fluigent, France), rinsed with mf-TIRF buffer (10 mM imidazole, pH 7.4, 50 mM KCl, 1 mM $MgCl_2$, 1 mM EGTA, 0.2 mM ATP, 10 mM DTT, and 1 mM DABCO) and incubated with 1% BSA and 10 µg/mL streptavidin in 20 mM HEPES pH 7.5, and 50 mM KCl for 5 min. About 100 pM Biotin-SNAP-mDia1 molecules in TIRF buffer were then flowed in and allowed to anchor on the glass coverslip. Actin filaments with barbed ends anchored to the formins were grown by flowing in a solution containing 1 µM G-actin (15% Alexa-488 labeled) and 0.5 µM profilin in mf-TIRF buffer. All experiments were carried out at room temperature in a TIRF buffer. Each experiment was repeated at least three times. Data from a single replicate is presented in the figures.

## Image acquisition and analysis

Single-wavelength time-lapse TIRF imaging was performed on a Nikon-Ti2000 inverted microscope equipped with a 40 mW Argon laser, a 60X TIRF-objective with a numerical aperture of 1.49 (Nikon Instruments Inc., USA), and an IXON LIFE 888 EMCCD camera (Andor Ixon, UK). One pixel was equivalent to 144 × 144 nm. Focus was maintained by the Perfect Focus system (Nikon Instruments Inc., Japan). Time-lapsed images were acquired every 2 or 5 s using Nikon Elements imaging software (Nikon Instruments Inc., Japan). For three-color images, the sample was sequentially excited by 488, 561, and 640 nm lasers. Images were acquired either continuously or with a 1 or 4 s delay between consecutive images.

Images were analyzed in Fiji[62]. Background subtraction was conducted using the rolling ball background subtraction algorithm (ball radius five pixels). Time-lapse images of between 50 and 100 filaments were acquired in a single field of view for each condition and all of these filaments were included to determine the cumulative distribution functions (CDFs) showing the time-dependent survival fraction of various complexes. For mf-TIRF assays, the kymograph plugin was used to draw kymographs of individual filaments. The kymographs were used to identify the timepoint of the detachment of filaments as a function of time. For three-color images, a 5 × 5-pixel box was drawn at the location of the barbed end of the filament and the time-dependent integrated intensity values were recorded for the single-molecule channels. The integrated intensity values were background corrected by subtracting the integrated intensity of a 5 × 5-pixel box drawn away from the filament.

Data analysis and curve fitting were carried out in Microcal Origin.

## Determination of rates of CP or formin dissociation from BFC complexes ($k'_F$ and $k'_C$)

The time-dependent fraction of BFC complexes dissociating by either transitioning to BF (re-elongation) or BC (detachment) upon addition of twinfilin to the flow were plotted versus time (black symbols in Fig. 2e). The kinetics of dissociation of BFC complex and simultaneous appearance of BF (magenta symbols) and BC (yellow symbols) were analyzed using the approach described in our earlier study[29]. Briefly, BFC dissociation can occur via one of the following two routes:

$$\mathrm{BFC} \xrightarrow{k'_{-C}} \mathrm{BF} + \mathrm{C} \tag{1}$$

$$\mathrm{BFC} \xrightarrow{k'_{-F}} \mathrm{BC} + \mathrm{F} \tag{2}$$

These reactions can be described by the following differential equations:

$$d(\mathrm{BFC})/dt = -\mathrm{BFC}(k'_{-F} + k'_{-C}) \tag{3}$$

$$d(\mathrm{BF})/dt = k'_{-C}(\mathrm{BFC}) \tag{4}$$

$$d(\mathrm{BC})/dt = k'_{-F}(\mathrm{BFC}) \tag{5}$$

The number of filaments transitioning out of BFC, and into BF and BC all vary exponentially with rate constant ($k'_F + k'_C$). The value of $k'_F$ and $k'_C$ was derived from the relative fraction of filaments in BF and BC states as follows.

$$\mathrm{BFC}(t)/\mathrm{BFC}_0 = e^{-(k'_{-C} + k'_{-F})t} \tag{6}$$

$$BF(t)/BFC_0 = \left(\frac{k'_{-C}}{k'_{-C} + k'_{-F}}\right) * \left(1 - e^{-(k'_{-C} + k'_{-F})t}\right) \qquad (7)$$

$$BC(t)/BFC_0 = \left(\frac{k'_{-F}}{k'_{-C} + k'_{-F}}\right) * \left(1 - e^{-(k'_{-C} + k'_{-F})t}\right) \qquad (8)$$

$BFC_0$ is the total number of filaments in the BFC state just prior to flowing in twinfilin. The ratio of the number of filaments taking either of the two routes (BF or BC) is given by $N_{BF}/N_{BC} = k'_{-C}/k'_{-F}$. Note that $k'_{-F} + k'_{-C}$ are observed rate constants as they depend upon the concentration of twinfilin as well as elongation rates of filaments associated with a formin.

### Statistical analysis and error bars for dissociation rates

The uncertainty in dissociation rates of formin bound to barbed ends (BF→B + F) or BFC complexes (BFC → BF + C or BFC → BC + F) were determined by bootstrapping strategy[34]. The dissociation rate was determined by fitting the survival fraction (or CDF) data to a single-exponential function ($y = e^{-kt}$ or $y = 1 - e^{-kt}$). A custom-written MATLAB code was then used to simulate BF (or BFC) complex lifetimes for N filaments (where N is the number of filaments in the particular experiment) based on the rate k determined from the experimental data. The simulation was repeated 1000 times to generate 1000 individual survival fractions of N filaments. Each dataset was then fit to an exponential function and an observed rate constant $k_{obs}$ was determined for each of the 1000 simulated datasets. The standard deviation of these estimated rates allowed us to determine the uncertainty in our measured rates.

### Reporting summary

Further information on research design is available in the Nature Portfolio Reporting Summary linked to this article.

## Data availability

Data supporting the findings of this manuscript are available from the corresponding author upon request. Source data are provided with this paper Source data are provided with this paper.

## Code availability

Code used in this manuscript is available from the corresponding author upon request.

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

## Acknowledgements

S.S. is grateful to Bruce Goode for many years of actin-related discussions and for his mentorship and teaching, not only in doing research but also in managing a lab. S.S. thanks Marie-France Carlier for introducing him to actin and Jeff Gelles for teaching him single-molecule imaging. S.S. also thanks his graduate advisors Vinod Subramaniam and Hans Kanger for their mentorship. We thank Jim bear for critical reading of the manuscript and feedback. We thank Pekka Lappalainen for generously sharing twinfilin plasmids and Jan Faix for sharing VASP chimera plasmids. We thank Ankita and Sandeep Choubey for their help with statistical analysis. We thank Sandeep Choubey, Ekram Towsif, and Surbhi Garg for their comments on the manuscript. We thank Matt Winfree, Thomas Pitta and Lauryn Luderman for their tireless support towards our microscopy efforts. This work was supported by NIH NIGMS grant R35GM143050 to S.S. and startup funds from Emory University to S.S.

## Author contributions

H.U. and I.G. conducted experiments and analyzed data. H.U. and S.S. prepared figures. S.S. designed experiments and supervised the project. S.S. and H.U. wrote the first draft of the manuscript and all authors contributed to the editing. S.S. acquired funding.

## Competing interests

The authors declare no competing interests.
