## [Peer Review File · Nature Communications]

REVIEWER COMMENTS

Reviewer #1 (Remarks to the Author):

Capping protein (CP), formins, and twinfilin are conserved regulators of actin filament barbed end dynamics. Previous studies demonstrated that formins and CP function antagonistically to promote and inhibit actin filament polymerization, respectively, but can associate with each other at filament barbed end through a mechanism where CP dissociates formin to terminate actin filament polymerization (Bombardier et al., 2015; Shekhar et al., 2015). Previous work also revealed that CP and twinfilin interact with each other, and that twinfilin dissociates CP from the filament barbed end to initiate filament depolymerization (Hakala et al., 2021; Mwangangi et al., 2021). Finally, an earlier publication (Shekhar et al., 2021) showed that twinfilin cannot inhibit formin-catalyzed actin filament barbed end polymerization.

Here, Ulrichs et al. studied the combined effects of CP, formin (Dia1), and twinfilin on actin filament barbed end dynamics. Based on elegant *in vitro* TIRF microscopy experiments, the authors propose that these three proteins form a transient complex, which controls the dynamics of actin filament barbed ends. Although the data presented in the manuscript are of excellent technical quality and convincing, the findings appear somewhat confirmatory. This is because the data presented here appear to demonstrate that twinfilin indeed uncaps actin filaments, and that the uncapped filament barbed ends can undergo formin-catalyzed elongation. Thus, it remains bit unclear if the 'trimeric complex' proposed here has any additional activities that have not been demonstrated for twinfilin, CP and formins in previous publications, and if the formin-twinfilin interplay also occurs in cells. Additionally, the manuscript contains some overstatements that should be deleted/edited to avoid confusion.

Specific comments:

1. The authors present beautiful data demonstrating that CP and formin can associate with each other at filament barbed ends, and that the arrival of CP inhibits filament polymerization. This is consistent with earlier publications by Bombardier et al., and Shekhar et al., (2015). They also demonstrate that arrival of twinfilin to a filament barbed end harboring CP and formin induces filament uncapping and subsequent formin-mediated polymerization. Again, this elegantly confirms the previous findings on the filament uncapping by twinfilin, and that twinfilin cannot inhibit formin-catalyzed filament assembly (Hakala et al., 2021; Shekhar et al., 2021). However, the possible novel activities of the trimeric complex on the filament barbed end dynamics remained obscure. Therefore, the authors should put their results better in light with previous publications by clearly explaining which results confirm earlier studies, and what are the really novel findings of the present study.

2. The authors state in the ‘abstract’ and in several instances throughout the manuscript text that ‘twinfilin acts as a pro-formin pro-polymerization factor’. This is misleading, because experiments presented in the manuscript do not provide any evidence of twinfilin increasing the actin filament assembly rate by formins, and the effects of twinfilin on formin processivity in the presence of CP can be explained by the previously-demonstrated filament uncapping activity of twinfilin. Thus, it appears that the data presented in the manuscript show that twinfilin uncaps filament barbed ends (as demonstrated by Hakala et al., 2021; Mwangangi et al., 2022) and that twinfilin does not inhibit formin-catalyzed filament barbed end assembly (this was already earlier shown in Shekhar et al., 2021). Thus, all statements about twinfilin acting as a pro-formin pro-polymerization factor should be deleted, because these are misleading and would just generate confusion among the readers.

3. In the ‘discussion’, the authors claim that twinfilin, CP and formin can simultaneously bind to actin filament barbed end. However, structural studies (Mwangangi et al., 2021; Funk et al., 2021 + several formin publications) show that the binding sites of these three proteins overlap with each other at terminal subunits of the filament barbed end. Thus, only one of the proteins can bind filament barbed end at the time, and hence these proteins most likely associate with each other through other mechanisms: twinfilin binds CP through its C-terminal tail and formin most likely associates with the capped barbed end through its filament side-binding activity or by sliding along the filament (see e.g. Bombardier et al., 2015). Thus, the manuscript text should be revised in the light of existing structural data.

4. The possible physiological role of twinfilin – formin interplay remains obscure. This is because cell biological studies provided evidence that twinfilin accelerates actin filament disassembly (rather than promotes filament assembly) in lamellipodia and streocilia (Peng et al., 2009; Hakala et al., 2021). Thus, the manuscript would be significantly stronger, if the authors could provide some evidence of twinfilin promoting formin-mediated actin filament assembly also in cells.

5. The authors ignored some important studies by others. For example, CP was shown to accelerate Arp2/3-catalyzed actin filament nucleation (e.g. Akin and Mullins, 2008; Funk et al., 2021), and this may be especially relevant for lamellipodial actin dynamics. The Funk et al. (2021) paper also presents the structure of CP bound to filament barbed end. Moreover, the effects of twinfilin on actin filament barbed end depolymerization in the presence of high concentrations of actin monomers were also shown by Hakala et al., (2021), but the authors only cite the Shekhar et al., (2021) publication for this activity. Finally, the dynamics of CP at filament barbed ends at the lamellipodium are controlled by CARMIL family proteins (e.g. Fujiwara et al., 2014) and membrane phosphoinositides (e.g. Schafer et al., 1996). Hence, twinfilin-catalyzed filament uncapping may not be critical for actin filament assembly at the plasma membrane, but instead twinfilin’s main role would be to promote filament disassembly as suggested by earlier studies on cells.

6. The three-color TIRF experiments presented in Fig. 3 are beautiful and very informative. It would be really interesting also simultaneously visualize CP and twinfilin (and perhaps also one of the twinfilin mutants), because such experiment may provide new insights into how twinfilin uncaps filament barbed ends.

7. As mentioned towards the end of 'discussion', studying the interplay between twinfilin (which appears to associate with the Arp2/3-nucleated actin networks in cells), CP and Ena/VASP would be interesting, and certainly increase the novelty of this study. This is because whereas formins and Arp2/3 appear to mainly function in different cellular actin filament structures, there is a significant amount of literature proposing interplay between Arp2/3, CP, and Ena/VASP.

Reviewer #2 (Remarks to the Author):

The paper presents single filament polymerization experiments to study how twinfilin and capping protein influence elongation of barbed ends associated with the formin mDia1. Previous work examined formin and CP as well as CP and twinfilin, but not the three together. The authors are experts doing these experiments and present some valuable new data. Fig 3 with observations of multiple single molecules interacting with barbed end is unique and informative. The "model" used to summarize the work is just a cartoon that does not consider the rates of the reactions, so it does not provide helpful insights.

I have many suggestions meant to be helpful to the authors about data analysis, interpretation of observations and especially about the presentation, which has many problems that can be fixed.

Abstract: This abstract is burdened by phrases such as "multiprotein ecosystem" but more importantly, it does not report the main results. Here I offer an alternative that does have the main results:

"We used microfluidics-assisted TIRF microscopy to investigate how formin mDia1, capping protein (CP) and twinfilin influence the elongation of actin filament barbed ends. As observed previously, association of CP with a growing actin filament end occupied by formin mDia1 stops elongation of the filament followed by dissociation of either CP or formin. Single molecule fluorescence showed that twinfilin does not bind actin filament ends occupied by formin mDia1 unless CP is present. The complex of the three proteins is transient, about 1 s, but the presence of twinfilin increases the rate of CP dissociation with resumption of actin filament elongation."

Results:

Experimental conditions: What is the rationale for the chosen concentrations of actin and profilin? The rate of formin dissociation depends on the rate of elongation (not mentioned), which in turn depends on the concentrations of actin monomers and profilin but these concentrations are not uniform across the experiments. The profilin concentration has a biphasic effect with an optimum. The profilin concentrations used with 1 μM actin in some experiments seem to be below the optima (about 4-5 μM) in other publications, so the actin may not be saturated with profilin. Is this intentional? The figure legends should state the concentrations of free actin and profilin-actin,

Data analysis: "Changes in survival fraction of formin-bound filaments were used to determine the dissociation rate constant ($k\text{-F}$) of formin from the barbed end." Here and throughout, these are actually k observed rather than rate constants, since the rate of dissociation depends on the rate of elongation, which is not taken into consideration.

Line 164: "The dissociation kinetics of BFC complexes into BC and BF were exponential with rate constant $k\text{-BFC} = k'\text{-F} + k'\text{-C}$ where $k'\text{-F}$ is the rate of dissociation of formin from the BFC complex and $k'\text{-C}$ is the rate of dissociation of CP from the BFC complex." Again, these are k observed, since they depend on the concentration of twf, which is not taken into account.

Lines 584-595: Since neither the elongation rate nor the twinfilin concentrations are considered, these equations are incomplete.

Line 813: "White arrowheads denote the moment of BFC dissociation." No, this is the moment that CP dissociates from BFC.

Line 816: "to determine BFC dissociation rate $k\text{-BFC}$." No, this is the moment that CP dissociates from BFC.

Line 817: "BFC dissociation rate." No, this is the moment that CP dissociates from BFC.

Fig 1G: What is the interpretation of this curve? Does this reflect CP binding?

Fig 2A: How do you know CP remains on the end of the dissociated filament?

Fig 2B upper panel: Does the rate increase with time?

Fig 2C: This graph makes no sense. The Y-axis label is "Decision complex survival" but the plots show the parameter increases with time. Therefore, it is not survival but more likely dissociation of CP and resumption of elongation over time. Add "Twinfilin" over the box of symbols.

Fig 2D: This is not BFC dissociation; it is CP dissociation from BFC and resumption of elongation + dissociation of F from the filament.

Fig 2F: This is really resumption of elongation rather than a direct measurement of CP dissociation.

Fig 3C: The elongation rate of ~30 subunits/s seems fast for 0.5 μM actin monomers and 1 μM profilin, about 7 times faster than expected for actin alone. On line 835, I think you mean to say for both (b) and (c) "showing CP binding to and stopping the elongation of the barbed end followed dissociation of a mDia1."

Fig 3H: The time scale should be the same as Fig 3G.

Discussion: The model in fig 5 and its discussion are unsatisfying, because the "model" is just a cartoon that does not take into account the quantitative aspects of the system. Neither the concentrations of the reactants nor the rates of the reactions are considered. The authors' presentation of rates in the results section was also qualitative. We read that "very rapid" is tens of seconds, whereby the reactions during actin filament elongation by formins, profilin and actin take place on a millisecond time scale. The authors should bring time and concentrations into their thinking and description of the reactions illustrated in Fig 5.

In addition, the authors should consider simulating models of the reactions in Figs 1 and 2, to see if they account for the time courses and the extends of the reactions.

Methods:

Actin: How does labeling affect polymerization \pm formins?

Line 397: pH 6.9 is a strange choice for labeling lysines.

Lines 425: Do the SNAP tags influence the activity of twf1, mDia1, CP? Are all the molecules labeled with one dye?

Line 459: How were both CP chains expressed? Only CP25 is mentioned.

Presentation

Title: The senior author already used “menage a trois” to describe formins and capping protein on the barbed end of an actin filament. Now we have four components; so this phrase is inconsistent with the previous use. Furthermore, some might object to the sexual overtones.

The Results section is very conversational with unnecessary history and material that belongs in the discussion. Just present the observations.

Poor choice of words. The following is a list of problems with the wording of the text:

Line 33: “relatively slowly growing” is inaccurate. At the leading edge, filaments grow at about 200 subunits per second.

Line 89: “monomers between (the) formin and the pre-existing fluorescent segment of the filament.” The incoming subunit binds to the end of the filament, which is not between the formin and the end.

Line 83: The first paragraph would be better as multiple paragraphs, each with a good topic sentence. Start the second paragraph with “Dissociation of the formin from the barbed end led to detachment...” Start the third paragraph with “We recorded the detachment of actin filaments in the field over time.”

Line 84 and elsewhere: “studied how CP and twinfilin influence formin’s processivity.” The authors misuse “processive and processivity:” they incorrectly call experiments on the dissociation of

components from barbed ends as experiments on processivity. One would never call the dissociation of a motor protein from its track processivity.

Line 97: "Fluorescent labeling of actin monomers diminishes formin's processivity³¹." What does this mean? Is this an effect on elongation or formin dissociation rate?

Line 101: "Renucleation experiments confirmed..." Renucleation is lab jargon; explain the experiment.

Line 108: "with varying amounts of CP." You mean range of concentrations of CP."

Line 109: "actin filaments continued to move along the flow..." You mean "away from the attached formin." The flow has nothing to do with this.

Line 124: "CP was sufficient to rapidly extinguish formin-induced actin assembly." Extinguish is not an appropriate word here. You mean "terminate." Note that this reaction takes tens of seconds, so it is not rapid compared with the rate of elongation.

Lines 135 and 350: "cellular organelles, such as filopodia, lamellipodia." These are not organelles. Line 355, linear bundles of actin filaments are not organelles.

Line 138: reword as "formin dissociates at a rate intermediate between that of control and capping protein..."

Line 142: "twinfilin's presence led to a rescue of the adverse effects of CP on formin's processivity." Rescue is not the best word to describe the observation; be explicit.

Line 150: What is the "decision complex?" This looks like bad lab jargon to me. Stick with BFC, which is clear and descriptive.

Line 152: "This caused an almost immediate arrest of formin's elongation." This is incorrect. The formin does not elongate; it's the barbed end.

Line 169: One of the most interesting observations is buried in the middle of a paragraph: “while 1 μ M twinfilin increased CP’s dissociation rate from BFC complex by \sim 11-fold, formin’s rate of dissociation did not change.”

Lines 187-200 belong in the discussion. Start the section with “To directly visualize the effects of twinfilin on the dynamics of BFC we used three-color single molecule TIRF imaging.”

Line 227: “About 73% of (the BFC) complexes (36 out of 49) transitioned to the 649-mDia1 state with 20 nM twinfilin...” Is 649-mDia1 state BF? If so, state this.

Lines 230-239: Belong in discussion.

Line 239: “destabilizes CP’s barbed end localization by binding side of actin filaments” needs a reference.

Paragraph starting with line 245: The topic sentence is buried at the end “Even at high twinfilin concentration, we never observed any barbed ends with 649-mDia1 and 549-mTwf1 jointly bound to it.”

Lines 264-267: Two sentences say the same thing.

Lines 270-271: “implying that twinfilin’s primary role might be in accelerating dissociation of the BFC complexes rather than in preventing their formation.” This does not make sense. What is the evidence against preventing formation?

Line 298: “CP on its own is capable of rapidly dissociating formin from [delete, formin-bound] barbed ends.” Should spell out what you mean by “rapid.”

Line 518 following: many problems with subscripts.

Line 768: Who is Ankita?

We thank the reviewers for their suggestions which have been extremely valuable in improving the manuscript. We are now submitting a revised version of the manuscript by taking into account reviewers' suggestions.

Reviewer #1

Capping protein (CP), formins, and twinfilin are conserved regulators of actin filament barbed end dynamics. Previous studies demonstrated that formins and CP function antagonistically to promote and inhibit actin filament polymerization, respectively, but can associate with each other at filament barbed end through a mechanism where CP dissociates formin to terminate actin filament polymerization (Bombardier et al., 2015; Shekhar et al., 2015). Previous work also revealed that CP and twinfilin interact with each other, and that twinfilin dissociates CP from the filament barbed end to initiate filament depolymerization (Hakala et al., 2021; Mwangangi et al., 2021). Finally, an earlier publication (Shekhar et al., 2021) showed that twinfilin cannot inhibit formin-catalyzed actin filament barbed end polymerization.

Here, Ulrichs et al. studied the combined effects of CP, formin (Dia1), and twinfilin on actin filament barbed end dynamics. Based on elegant in vitro TIRF microscopy experiments, the authors propose that these three proteins form a transient complex, which controls the dynamics of actin filament barbed ends. Although the data presented in the manuscript are of excellent technical quality and convincing, the findings appear somewhat confirmatory. This is because the data presented here appear to demonstrate that twinfilin indeed uncaps actin filaments, and that the uncapped filament barbed ends can undergo formin-catalyzed elongation. Thus, it remains bit unclear if the 'trimeric complex' proposed here has any additional activities that have not been demonstrated for twinfilin, CP and formins in previous publications, and if the formin-twinfilin interplay also occurs in cells. Additionally, the manuscript contains some overstatements that should be deleted/edited to avoid confusion.

We thank the reviewer for the kind words about our previous work and appreciation for our TIRF experiments. While we partially agree with the reviewer that some of the data presented is confirmatory, we have here reported a number of novel discoveries. To name a few:

1. Our study is the first ever to show that a polymerase, a depolymerase and a blocker can simultaneously associate with the barbed end of an actin filament.
2. The reviewer rightly points out that we have previously shown that twinfilin doesn't inhibit formin-catalyzed barbed end polymerization. We clarify that previously, only the effects of twinfilin on formin elongation rate had been investigated. Our current study is the first ever investigation of twinfilin's effects on formin's processivity. We find twinfilin doesn't bind actin filament ends occupied by formin unless CP is present.
3. The reviewer also correctly points out that CP and twinfilin interact with each other (in solution), and that twinfilin dissociates CP from barbed ends. However, there is no evidence that twinfilin's interaction with CP is necessary for its ability to uncap. On the contrary, Hakala et al (NCB, 2021) showed that mutations in twinfilin's tail which are thought to disrupt its binding to CP, don't inhibit twinfilin's uncapping. This mutant actually leads to much higher uncapping than wild-type twinfilin.

4. Based on the reviewer's suggestion, we have now conducted additional experiments to directly visualize CP and twinfilin at the actin barbed end. We find that twinfilin is a very weak uncapper and on average it takes about 30 individual interactions of twinfilin with CP-bound barbed ends to cause CP's removal.
5. In contrast, when formin and CP are both present, twinfilin's exhibits a drastic increase in its uncapping efficiency. Only a single interaction of twinfilin molecule with the barbed end is sufficient to dissociate CP and facilitate resumption of formin-based elongation.

Specific comments:

1. The authors present beautiful data demonstrating that CP and formin can associate with each other at filament barbed ends, and that the arrival of CP inhibits filament polymerization. This is consistent with earlier publications by Bombardier et al., and Shekhar et al., (2015). They also demonstrate that arrival of twinfilin to a filament barbed end harboring CP and formin induces filament uncapping and subsequent formin-mediated polymerization. Again, this elegantly confirms the previous findings on the filament uncapping by twinfilin, and that twinfilin cannot inhibit formin-catalyzed filament assembly (Hakala et al., 2021; Shekhar et al., 2021). However, the possible novel activities of the trimeric complex on the filament barbed end dynamics remained obscure. Therefore, the authors should put their results better in light with previous publications by clearly explaining which results confirm earlier studies, and what are the really novel findings of the present study.

Like we have pointed out earlier, while the reviewer is partially correct that we had, in our 2021 study, looked into twinfilin's effect on formin, we only investigated twinfilin's effect on formin's elongation rate. Our current manuscript is the first study to quantitatively show that twinfilin has no effect on formin's dissociation from the barbed end. We would further like to stress that just because a protein does not change formin's elongation rate (which was published before) in no way implies that it cannot affect its barbed end residence time. Further, in contrast with reviewer's statement, there is evidence that formin and CP do not associate with each other at barbed ends. They both bind to the barbed end but do not bind each other at the barbed end.

Nevertheless, we agree with the reviewer that we could have done a better job in distinguishing our novel findings from previously published results. Therefore, we have now added the following text about previous studies to better contextualize our discoveries.

"Although the individual effects of formin, CP and twinfilin on actin assembly are relatively well characterized, how they simultaneously act in multiprotein teams at barbed ends to influence actin assembly is still poorly understood. This question is especially important as these factors are often found in the same cellular compartments such as filopodia, lamellipodia and stereocilia (Avenarius et al., 2017; Block et al., 2012; Peng et al., 2009; Shekhar et al., 2021; Vartiainen et al., 2003; Yang et al., 2007). In addition to twinfilin's monomer sequestration and barbed end depolymerization functions, it can directly bind CP via its C-terminal tail and accelerate CP's removal from barbed ends, ultimately reducing CP's barbed end dwell time by about 6-fold (Hakala et al., 2021; Mwangangi et al., 2021). The CP-twinfilin interaction has been demonstrated to be required for proper intracellular localization of CP but is dispensable for twinfilin's uncapping in vitro (Hakala et al., 2021; Myers et al., 2022). Moreover, in absence of

direct visualization of CP's uncapping by twinfilin, the underlying mechanism remains unclear. While a number of recent studies have looked at this twinfilin-CP interaction, twinfilin's effects on formin have not been extensively studied. We previously reported that formin's rate of barbed end elongation is not affected by high concentration of twinfilin (Shekhar et al., 2021). Nevertheless, twinfilin's effects on formin's long-lived residence at the barbed end has yet to be investigated."

2. The authors state in the 'abstract' and in several instances throughout the manuscript text that 'twinfilin acts as a pro-formin pro-polymerization factor'. This is misleading, because experiments presented in the manuscript do not provide any evidence of twinfilin increasing the actin filament assembly rate by formins, and the effects of twinfilin on formin processivity in the presence of CP can be explained by the previously-demonstrated filament uncapping activity of twinfilin. Thus, it appears that the data presented in the manuscript show that twinfilin uncaps filament barbed ends (as demonstrated by Hakala et al., 2021; Mwangangi et al., 2022) and that twinfilin does not inhibit formin-catalyzed filament barbed end assembly (this was already earlier shown in Shekhar et al., 2021). Thus, all statements about twinfilin acting as a pro-formin pro-polymerization factor should be deleted, because these are misleading and would just generate confusion among the readers.

We respectfully disagree with the reviewer on this point. We fully agree with the reviewer that "twinfilin doesn't increase the assembly rate of formins". We have never made this claim either. However, our experiments clearly show that in presence of capping protein (50 nM), there is a 10-fold reduction in formin's processivity. When twinfilin is added, the same concentration of CP only leads to a 3-fold reduction in formin's processivity. This means that addition of twinfilin leads to longer formin dwell times at the barbed which in turn will result in longer filaments i.e. increased polymerization by formin.

Having said that, we agree with the reviewer that this claim can be toned down. As requested by the reviewer, all mentions of "pro-elongation" have been removed. We have additionally removed all but one mention of the phrase "pro-formin factor". We would still like to keep this phrase in the abstract.

"Thus, the depolymerase twinfilin acts as a pro-formin factor that promotes polymerization when both CP and formin are present."

3. In the 'discussion', the authors claim that twinfilin, CP and formin can simultaneously bind to actin filament barbed end. However, structural studies (Mwangangi et al., 2021; Funk et al., 2021 + several formin publications) show that the binding sites of these three proteins overlap with each other at terminal subunits of the filament barbed end. Thus, only one of the proteins can bind filament barbed end at the time, and hence these proteins most likely associate with each other through other mechanisms: twinfilin binds CP through its C-terminal tail and formin most likely associates with the capped barbed end through its filament side-binding activity or by sliding along the filament (see e.g. Bombardier et al., 2015). Thus, the manuscript text should be revised in the light of existing structural data.

We respectfully disagree with the view that only one of these proteins can bind a barbed end at a time. As the reviewer points out, all of the mentioned insights have been derived from studies

looking at interactions of only one of these proteins at a time with actin. Moreover, these studies show that each of these proteins have multiple different contact sites with the two terminal actin subunits of the barbed end. As a result, when two or more proteins are bound to the actin barbed end, they might not be able to access all of their binding sites on the terminal actin subunits. Thus, leading to lowered individual affinities for the actin filament barbed end.

For formin – CP simultaneous binding, we had proposed a structural model in our early study (Shekhar et al, 2015, Nat. Comms) which would allow both formin and CP to simultaneously bind the barbed end. By superimposing the individual structures of the two proteins, we found that the main clash occurs between the B-tentacle of CP and the post region of formin FH2 for binding the terminal subunit. Previous studies have shown that removal of the β -tentacle doesn't abolish CP's barbed end binding but only causes a reduction in its affinity for barbed ends. As a result, prevention of CP's β -tentacle from accessing its binding site on actin by formin is expected to cause CP's faster dissociation from barbed ends when formin is also present. This is exactly what we observed experimentally, CP's affinity was greatly reduced when formin was already present at the barbed end (Shekhar et al, 2015, Nat. Comms). We believe these observations provide one possible configuration for them to simultaneously bind a barbed end.

For CP-twinfilin simultaneous binding, mTwf1's tail and CP's β -tentacle are found in close vicinity of each other in the pocket of the terminal actin subunit (Mwangangi et al., 2021, Science Advances). CP and twinfilin are known to interact with each other via the CPI-motif binding site of CP and twinfilin's c-terminal tail (Johnston et al., 2018, eLife). However, in the published crystal structure of CP-twinfilin bound to two actin subunits in Mwangangi et al., 2021, the CPI-motif binding regions of CP are not in proximity of the twinfilin tail – suggesting the two proteins are not interacting with each other when bound to actin, at least not via their canonical binding sites.

In summary, based off these structural insights, we believe it is a leap to conclude that there is no way that two or more of these proteins can simultaneously bind the same barbed end.

4. The possible physiological role of twinfilin – formin interplay remains obscure. This is because cell biological studies provided evidence that twinfilin accelerates actin filament disassembly (rather than promotes filament assembly) in lamellipodia and streocilia (Peng et al., 2009; Hakala et al., 2021). Thus, the manuscript would be significantly stronger, if the authors could provide some evidence of twinfilin promoting formin-mediated actin filament assembly also in cells.

We completely agree with the reviewer that conclusions presented here will be greatly strengthened by cellular evidence of twinfilin promoting formin-mediated actin assembly in cells. We have reached out to two labs with expertise in cell biological studies of actin dynamics to carry out localized (in)activation of twinfilin in B16 cells using optogenetic tools to investigate its effects on formin-associated actin structures. However, as we hope the reviewer will appreciate, cellular studies will, at the very least, require years of additional work. This would be an entirely new project, which we feel is outside the scope of the current study.

Nevertheless, the following sentence in the discussion section stresses the importance of these future experiments:

“Future cellular experiments where twinfilin activity can be locally enhanced/suppressed by optogenetic techniques are needed to test these predictions.”

5. The authors ignored some important studies by others. For example, CP was shown to accelerate Arp2/3-catalyzed actin filament nucleation (e.g. Akin and Mullins, 2008; Funk et al., 2021), and this may be especially relevant for lamellipodial actin dynamics. The Funk et al. (2021) paper also presents the structure of CP bound to filament barbed end. Moreover, the effects of twinfilin on actin filament barbed end depolymerization in the presence of high concentrations of actin monomers were also shown by Hakala et al., (2021), but the authors only cite the Shekhar et al., (2021) publication for this activity. Finally, the dynamics of CP at filament barbed ends at the lamellipodium are controlled by CARMIL family proteins (e.g. Fujiwara et al., 2014) and membrane phosphoinositides (e.g. Schafer et al., 1996). Hence, twinfilin-catalyzed filament uncapping may not be critical for actin filament assembly at the plasma membrane, but instead twinfilin’s main role would be to promote filament disassembly as suggested by earlier studies on cells.

Thank you for pointing this out – our apologies for missing these key references. We have now added two more citations of Hakala et al. 2021 and Johnston et al, 2015 as evidence for twinfilin’s depolymerizing function in presence of actin monomers. We have further added the following text to the manuscript taking into account the role of CARMIL, V-1/myotrophin and phosphoinositides.

“In the future, it will be interesting to explore if other factors that bind either CP, formin, twinfilin or actin barbed ends, like CARMIL(Edwards et al., 2014; Fujiwara et al., 2014), V-1/myotrophin(Fujiwara et al., 2014), APC(Breitsprecher et al., 2012), Spire(Montaville et al., 2014), IQGAP(Hoeprich et al., 2022) and polyphosphoinositides(Hakala et al., 2018; Schafer et al., 1996), might also form their own multiprotein ecosystems with these proteins at the barbed ends to tune actin assembly.”

While the studies from Funk et al and Akin et al have made valuable contributions to our understanding of the role of CP in assembling branched actin networks, we have elected to not cite these papers since our current study is primarily focused on formin-assembled filaments.

6. The three-color TIRF experiments presented in Fig. 3 are beautiful and very informative. It would be really interesting also simultaneously visualize CP and twinfilin (and perhaps also one of the twinfilin mutants), because such experiment may provide new insights into how twinfilin uncaps filament barbed ends.

We thank the reviewer for their kinds words for our single-molecule experiments and encouraging us to conduct experiments to simultaneously visualize CP and twinfilin.

As suggested by the reviewer, we have now conducted extensive three-color experiments to simultaneously visualize CP and twinfilin at filament barbed ends. The results are presented in the new figure 4. Using labelled CP and labelled twinfilin, we observed that twinfilin transiently associates with CP-bound barbed ends (Figure 4a,b). However, owing to the slow rate of uncapping by twinfilin (it only accelerates uncapping rate by 6-fold) and transient nature of these interactions, these experiments required long-durations with frequent observation (several frames per second) of CP-bound filaments which caused excessive photobleaching of labelled CP.

Therefore, for quantitative characterization of uncapping by twinfilin, we recorded the arrival and departure of labelled twinfilin molecules on filament barbed ends bound to unlabeled CP (Figure 4c-e). CP's departure was identified by filament depolymerization (i.e. due to loss of CP). We found that successful removal of CP takes on average about 30 rounds of binding and unbinding of twinfilin molecules and each of these colocalization events lasts about 1.9 ± 0.1 seconds.

7. As mentioned towards the end of 'discussion', studying the interplay between twinfilin (which appears to associate with the Arp2/3-nucleated actin networks in cells), CP and Ena/VASP would be interesting, and certainly increase the novelty of this study. This is because whereas formins and Arp2/3 appear to mainly function in different cellular actin filament structures, there is a significant amount of literature proposing interplay between Arp2/3, CP, and Ena/VASP.

These are great points. Incorporating VASP into our multiprotein experiments is a great idea and we thank the reviewer for bringing this up. However, these experiments have been challenging to conduct within the timeframe of the revision for the following reasons. Firstly, when compared to formin mDia1, which remains bound to barbed ends for several tens of minutes, Ena/VASP proteins exhibit much shorter barbed end residence. *Drosophila* Enabled (Ena) remains bound to filament ends for only about 10 s (Winkelman et al, 2014). In addition, (Mammalian) VASP elongate actin filaments poorly due to their low affinity G-actin binding site (GAB) (Breitsprecher et al., 2011, EMBO J). Further, barbed dwell times of these proteins are greatly affected by whether measurements are made in solution or on surfaces, further complicating the experiments. With help from Jan Faix, we are therefore currently exploring human VASP chimeras containing the GAB domain from *Dictyostelium* which are known to have a much higher G-actin affinity (Breitsprecher et al., 2011, EMBO J). As a result, these experiments will require many more months (if not years) of additional work and will be followed up in a follow-up study.

Nevertheless, the following sentence in the discussion section stresses the importance of these future experiments:

“Although, it remains to be tested if twinfilin also favors Ena/VASP over CP and acts as a pro-Ena/VASP factor. Future cellular experiments where twinfilin activity can be locally enhanced/suppressed by optogenetic techniques are needed to test these predictions.”

Reviewer #2 (Remarks to the Author):

1. The paper presents single filament polymerization experiments to study how twinfilin and capping protein influence elongation of barbed ends associated with the formin mDia1. Previous work examined formin and CP as well as CP and twinfilin, but not the three together. The authors are experts doing these experiments and present some valuable new data. Fig 3 with observations of multiple single molecules interacting with barbed end is unique and informative. The “model” used to summarize the work is just a cartoon that does not consider the rates of the reactions, so it does not provide helpful insights.

We thank the reviewer for their appreciation of our work and for the detailed feedback/advice that has helped us make the manuscript more accurate and improve the presentation.

While we agree that the “model” would be made much more powerful by taking into account reaction rates, we currently don’t have access to all the rates needed to build a comprehensive and accurate quantitative model of how the three proteins co-regulate barbed end dynamics. Our purpose of presenting this figure is rather to attempt to integrate our discoveries in the context of the current knowledge of the field. Nevertheless, in view of reviewer’s thoughts we have elected to replace the word “model” with “schematic”.

2. I have many suggestions meant to be helpful to the authors about data analysis, interpretation of observations and especially about the presentation, which has many problems that can be fixed.

Thank you, we appreciate your thoughtful suggestions.

3. Abstract: This abstract is burdened by phrases such as “multiprotein ecosystem” but more importantly, it does not report the main results. Here I offer an alternative that does have the main results:

“We used microfluidics-assisted TIRF microscopy to investigate how formin mDia1, capping protein (CP) and twinfilin influence the elongation of actin filament barbed ends. As observed previously, association of CP with a growing actin filament end occupied by formin mDia1 stops elongation of the filament followed by dissociation of either CP or formin. Single molecule fluorescence showed that twinfilin does not bind actin filament ends occupied by formin mDia1 unless CP is present. The complex of the three proteins is transient, about 1 s, but the presence of twinfilin increases the rate of CP dissociation with resumption of actin filament elongation.”

We thank the reviewer for suggesting improvements to the abstract. We have now re-written the abstract taking the reviewer’s advice into consideration. Please see below for the text of the new abstract

“Living cells assemble their actin networks by regulating reactions at the barbed end of actin filaments. Formins accelerate elongation, capping protein (CP) arrests growth and twinfilin promotes depolymerization at barbed ends. How cells integrate these disparate activities within a shared cytoplasm to produce diverse actin networks, each with distinct morphologies and finely tuned assembly kinetics, is unclear. We used microfluidics-assisted TIRF microscopy to investigate how formin mDia1, CP and twinfilin influence the elongation of actin filament barbed ends. We discovered that the three proteins can simultaneously bind a barbed end in a multiprotein complex. Three-color single molecule experiments showed that twinfilin cannot bind actin filament ends occupied by formin mDia1 unless CP is present. The trimeric complex is short-lived (~1s) and results in rapid dissociation of CP by twinfilin causing resumption of rapid formin-based elongation. Thus, the depolymerase twinfilin acts as a pro-formin factor that promotes polymerization when both CP and formin are present. While a single twinfilin binding event is sufficient to displace CP from the trimeric complex, it takes about 30 independent twinfilin binding events to remove capping protein from CP-bound barbed end. Our findings establish a new paradigm in which polymerases, depolymerases and cappers work in concert to tune cellular actin assembly.”

Results:

4. Experimental conditions: What is the rationale for the chosen concentrations of actin and profilin? The rate of formin dissociation depends on the rate of elongation (not mentioned), which in turn depends on the concentrations of actin monomers and profilin but these concentrations are not uniform across the experiments. The profilin concentration has a biphasic effect with an optimum. The profilin concentrations used with 1 μM actin in some experiments seem to be below the optima (about 4-5 μM) in other publications, so the actin may not be saturated with profilin. Is this intentional? The figure legends should state the concentrations of free actin and profilin-actin.

Indeed, the use of different concentrations of profilin and actin across an experiment is intentional. Formins do not nucleate efficiently when all the actin subunits are bound to profilin— therefore for the nucleation step of our experiments we used 1 μM G-actin and 0.5 μM profilin. To determine if the filament was indeed elongating from the coverslip-anchored formin (and that the filament was not stuck on the surface via its pointed end), the filaments were transiently exposed to a flow of 1 μM unlabeled G-actin and 4 μM profilin which transiently increased elongation rate and allowed ascertaining that the filaments were indeed growing from coverslip-anchored formins. However, rapid elongation by formin leads to lengthening of filaments which in turn accelerates their detachment from surface-anchored formins due to increased drag force (Cao et al, 2018, eLife). To reduce the contribution of pulling force in formin's dissociation, we therefore used 0.2 μM unlabeled G-actin with 0.7 μM profilin in the following step – this concentration of profilin and actin allows visible elongation by formin but not enough that the pulling force on these filaments caused their detachment.

We have elected not to specifically mention concentrations of free actin and profilin-actin is this can be calculated by informed readers based on protein concentrations provided in the legend.

5. Data analysis: “Changes in survival fraction of formin-bound filaments were used to determine the dissociation rate constant ($k\text{-F}$) of formin from the barbed end.” Here and throughout, these are actually k observed rather than rate constants, since the rate of dissociation depends on the rate of elongation, which is not taken into consideration.

We have now inserted the following text to make this point:

*“Changes in survival fraction of formin-bound filaments were then used to determine the **observed** dissociation rate of formin from the barbed end.”*

6. Line 164: “The dissociation kinetics of BFC complexes into BC and BF were exponential with rate constant $k\text{-BFC} = k'\text{-F} + k'\text{-C}$ where $k'\text{-F}$ is the rate of dissociation of formin from the BFC complex and $k'\text{-C}$ is the rate of dissociation of CP from the BFC complex.” Again, these are k observed, since they depend on the concentration of twf, which is not taken into account.

We have now edited this sentence as follows to make it clear that these are “observed” rates :

*“The dissociation kinetics of BFC complexes into BC and BF were exponential with **observed** rate $k\text{-BFC} = k'\text{-F} + k'\text{-C}$ where $k'\text{-F}$ is the **observed** rate of dissociation of formin from the BFC complex and $k'\text{-C}$ is the **observed** rate of dissociation of CP from the BFC complex”*

7. Lines 584-595: Since neither the elongation rate nor the twinfilin concentrations are considered, these equations are incomplete.

The following line has now been added at the end of the paragraph to make it clear that these rates depend upon twinfilin concentration and elongation rate.

“Note that $k'_{-F} + k'_{-C}$ are observed rates as they depend upon the concentration of twinfilin as well as elongation rate of formins.”

Further, “Dissociation rate of CP (k'_{-c}) or Formin (k'_{-f}) from BFC (s^{-1})” has been replaced by “Observed dissociation rate of CP (k'_{-c}) or Formin (k'_{-f}) from BFC (s^{-1})” in the Y-axis label of Fig. 2f.

8. Line 813: “White arrowheads denote the moment of BFC dissociation.” No, this is the moment that CP dissociates from BFC.

Thank you for pointing this out. This sentence has now been changed as follows:

“White arrowheads denote the moment formin or CP dissociation from the BFC complex.”

9. Line 816: “to determine BFC dissociation rate k_{-BFC} .” No, this is the moment that CP dissociates from BFC.

Thank you for pointing this out. This sentence has now been changed as follows

“to determine the rate of BFC disassembling into BF or BC.”

10. Line 817: “BFC dissociation rate.” No, this is the moment that CP dissociates from BFC.

“BFC dissociation rate.” has now been changed to “Rate of BFC dissociation into BC or BF”

11. Fig 1G: What is the interpretation of this curve? Does this reflect CP binding?

No, this represents the rate at which formin dissociates from the barbed end. To clarify, while every formin-bound filament would get paused upon CP binding ($BF + C \rightarrow BFC$), not all of these filaments will immediately get detached. About half of the BFC filaments would resume elongation due to CP's dissociation (returning to the BF state, $BFC \rightarrow BF + C$), and the other half would detach due to formin's detachment ($BFC \rightarrow BC + F$). The BF filaments would then undergo another round of pausing by CP binding and the cycle would repeat itself until all the filaments get detached from formins.

12. Fig 2A: How do you know CP remains on the end of the dissociated filament?

Thank you for pointing this out. From these experiments we don't. However an earlier study by Bombardier et al (2015) from the labs Jeff Gelles and Bruce Goode using labelled CP and formin had found that upon dissociation of formin, the CP remained bound to the barbed end. We have repeated these experiments in our current study and have made similar observations (Fig. 3).

13. Fig 2B upper panel: Does the rate increase with time?

No. However it would have, had we used a higher concentration of profilin-bound actin monomers and faster flow rates. Nevertheless, to completely eliminate the effect of flow/force, we repeated these experiments with labelled proteins using conventional non-microfluidics TIRF imaging (Fig. 3). The two approaches gave us similar results.

14. Fig 2C: This graph makes no sense. The Y-axis label is “Decision complex survival” but the plots show the parameter increases with time. Therefore, it is not survival but more likely dissociation of CP and resumption of elongation over time. Add “Twinfilin” over the box of symbols.

The Y-axis label was a typing mistake and has been corrected to “*Fraction of BFC complexes disassembled*”. As advised, we have also added “mTwf1” in the box with symbols. Thank you for picking up on these.

15. Fig 2D: This is not BFC dissociation; it is CP dissociation from BFC and resumption of elongation + dissociation of F from the filament.

No, this is a combination of BFC disassembling via either of the two routes i.e. by dissociation of CP causing a resumption of elongation, or via dissociation of formin causing the disappearance of CP-bound filament in the flow.

16. Fig 2F: This is really resumption of elongation rather than a direct measurement of CP dissociation.

Yes the reviewer is correct but resumption of elongation can only occur due to CP dissociation.

17. Fig 3C: The elongation rate of ~30 subunits/s seems fast for 0.5 μM actin monomers and 1 μM profilin, about 7 times faster than expected for actin alone. On line 835, I think you mean to say for both (b) and (c) “showing CP binding to and stopping the elongation of the barbed end followed dissociation of a mDia1.”

The reviewer is correct. The rate of elongation for formin bound filaments in 3c and 3d are slightly faster than what is expected for 0.5 μM actin monomers and 1 μM profilin. We believe this is either due to overestimation of stock concentration of actin used in these particular experiments or a pipetting error. Nevertheless, this doesn't change the conclusions we draw from this figure.

18. Fig 3H: The time scale should be the same as Fig 3G.

Thank you for pointing this out. We have now fixed this.

19. Discussion: The model in fig 5 and its discussion are unsatisfying, because the “model” is just a cartoon that does not take into account the quantitative aspects of the system. Neither the concentrations of the reactants nor the rates of the reactions are considered. The authors' presentation of rates in the results section was also qualitative. We read that “very rapid” is tens of seconds, whereby the reactions during actin filament elongation by formins, profilin and actin take place on a millisecond time scale. The authors should bring time and concentrations into their thinking and description of the reactions illustrated in Fig 5.

In addition, the authors should consider simulating models of the reactions in Figs 1 and 2, to see if they account for the time courses and the extends of the reactions.

As we have mentioned earlier, we agree that fig 6 (Fig 5 in the reviewed version of the manuscript) would be made much stronger by taking into account reaction rates. However, we currently don't have access to all the rates needed to build a comprehensive and accurate quantitative model of how the three proteins co-regulate barbed end dynamics. Our purpose of presenting this figure is to contextualized our discoveries. We understand that the word “model” can mean different things to

different people. In absence of quantitative numbers, we have elected to replace the word “model” with “schematic”.

Thank you for this recommendation to simulate these reactions. We have already initiated a collaboration with a theorist with whom we are taking a deeper look in to this. This will be part of a follow-up study.

Methods:

20. Actin: How does labeling affect polymerization \pm formins?

An earlier study from the labs of Guillaume Romet-Lemonne and Antoine Jegou (Cao et al, 2018 eLife) conducted a detailed investigation of formin as a function of actin labelling fraction. They found that actin labelling slowed down the elongation of formin filaments and significantly accelerated formin dissociation rate. This is the reason why we only used labelled actin for nucleation of the pointed-end fragment of the filament in our mf-TIRF experiments (Fig. 1a and 2a). Unlabeled actin was used for the elongation step.

21. Line 397: pH 6.9 is a strange choice for labeling lysines.

pH 6.9 was used to label actin in the Carrier lab where the PI had trained as a postdoc.

These reactions are indeed more efficient at higher pH, in the range of 7.5 – 8.5. However, for labeling lysines on actin with Alexa-488 succinimidyl ester dye, we haven't seen any major difference in labeling efficiency between labelling at pH 6.9 or at pH 7.8.

22. Lines 425: Do the SNAP tags influence the activity of twf1, mDia1, CP? Are all the molecules labeled with one dye?

SNAP-tagged formin and CP constructs used here have previously been employed in a number of earlier studies (Bombardier et al, 2015 Nat. Comms, Breitsprecher et al 2012 Science). Appropriate controls were conducted to ensure that SNAP tag did not visibly affect activities of these proteins (see supplementary fig 3 in Bombardier et al, 2015).

Ours is the first study using the mouse SNAP-tagged twinfilin-1 construct. We confirmed that twinfilin's uncapping activity was not altered by addition of the SNAP tag (please see supplementary figure 5).

We now also present photobleaching tests on these labelled proteins. Please see supplementary figures 3, 4 and 6. We find that most SNAP-CP and SNAP-mTwf1 molecules are bound to a single dye molecule. About 28% of SNAP-mDia1 are labelled with 2 dye molecules and 68% SNAP-mDia1 are labeled with a single dye molecule.

23. Line 459: How were both CP chains expressed? Only CP25 is mentioned.

Capping protein α 1 and β 2 were expressed from the same plasmid with a single His-tag on the alpha subunit. SNAP-CP was expressed from a single plasmid containing His- and SNAP-tagged β 1 subunit and untagged α 1. This information has now been added to the methods section of the manuscript.

Presentation

24. Title: The senior author already used “menage a trois” to describe formins and capping protein on the barbed end of an actin filament. Now we have four components; so this phrase is inconsistent with the previous use. Furthermore, some might object to the sexual overtones.

The title has now been changed to *“Multicomponent regulation of actin barbed end assembly by twinfilin, formin and capping protein”*

25. The Results section is very conversational with unnecessary history and material that belongs in the discussion. Just present the observations.

The results section has now been edited as suggested and parts of it have been either moved to the discussion or deleted entirely.

26. Poor choice of words. The following is a list of problems with the wording of the text:

Line 33: “relatively slowly growing” is inaccurate. At the leading edge, filaments grow at about 200 subunits per second.

We disagree, this phrase is comparing the rate of elongation of Arp2/3 nucleated free barbed ends with that of formin-bound barbed ends. The word “relatively” clarifies that we are not talking about absolute numbers, just comparing formin-based elongation with free barbed end elongation at the same profilin and actin concentration.

27. Line 89: “monomers between (the) formin and the pre-existing fluorescent segment of the filament.” The incoming subunit binds to the end of the filament, which is not between the formin and the end.

Thank you. This has now been corrected. And the text has been changed to

“Since elongation occurs by insertion of unlabeled monomers at the formin bound barbed end, the pre-existing fluorescent segment of the filament appears to move in the direction of the flow (Fig. 1a, b; Supplementary movie 1), away from the location of formin anchoring.”

28. Line 83: The first paragraph would be better as multiple paragraphs, each with a good topic sentence. Start the second paragraph with “Dissociation of the formin from the barbed end led to detachment...” Start the third paragraph with “We recorded the detachment of actin filaments in the field over time.”

Done.

29. Line 84 and elsewhere: “studied how CP and twinfilin influence formin’s processivity.” The authors misuse “processive and processivity:” they incorrectly call experiments on the dissociation of components from barbed ends as experiments on processivity. One would never call the dissociation of a motor protein from its track processivity.

We are not calling dissociation of formin from the barbed end as its processivity. However, the two terms are related i.e. increase in dissociation of formin from the barbed end does lead to lower processivity. Processivity is a commonly used term to describe formin’s dwell times at the barbed

end in a number of earlier studies including ours (please see Schmidt et al, 2021, PNAS; Shekhar et al, 2105, Nat. Comms.; Cao et al, 2018, eLife; Romero et al, 2007, JBC).

We have now completely removed the following sentence.

“In addition, our mf-TIRF experimental strategy allows processivity measurements of filaments elongating from unlabeled actin monomers - thus eliminating the influence of labeled actin on formin’s processivity”

30. Line 97: “Fluorescent labeling of actin monomers diminishes formin’s processivity³¹.” What does this mean? Is this an effect on elongation or formin dissociation rate?

Both. Please see Cao et al, 2018, eLife. However, to reduce confusion, we have now removed this sentence

31. Line 101: “Renucleation experiments confirmed...” Renucleation is lab jargon; explain the experiment.

This sentence has now been replaced with:

“To confirm that the filament disappearance was due to detachment of filaments from the formin rather than due to detachment of the entire formin-filament complex from the glass coverslip, we re-exposed the surface to a flow containing 1 μ M Alexa-488 G-actin and 0.5 μ M profilin.”

32. Line 108: “with varying amounts of CP.” You mean range of concentrations of CP.”

The phrase “varying amounts” has now been replaced with “varying concentrations” throughout the text.

33. Line 109: “actin filaments continued to move along the flow...” You mean “away from the attached formin.” The flow has nothing to do with this.

This has now been reworded as follows:

“In control reactions, the fluorescent segment of actin filaments continued to move away from the attached formin, at a constant speed, in the direction of the flow, indicating processive elongation by formin.”

34. Line 124: “CP was sufficient to rapidly extinguish formin-induced actin assembly.” Extinguish is not an appropriate word here. You mean “terminate.” Note that this reaction this takes tens of seconds, so it is not rapid compared with the rate of elongation.

The sentence has now been reworded to:

“This means that even low concentrations of active CP were sufficient to rapidly terminate formin-induced actin assembly as compared to control.”

35. Lines 135 and 350: “cellular organelles, such as filopodia, lamellipodia.” These are not organelles. Line 355, linear bundles of actin filaments are not organelles.

The word “organelles” has been replaced with “compartments” or “structures”.

36. Line 138: reword as “formin dissociates at a rate intermediate between that of control and capping protein...”

Done.

37. Line 142: “twinfilin’s presence led to a rescue of the adverse effects of CP on formin’s processivity.” Rescue is not the best word to describe the observation; be explicit.

“Rescue” has been replaced by “reduction”.

38. Line 150: What is the “decision complex?” This looks like bad lab jargon to me. Stick with BFC, which is clear and descriptive.

Done.

39. Line 152: “This caused an almost immediate arrest of formin’s elongation.” This is incorrect. The formin does not elongate; it’s the barbed end.

“Formin’s elongation” has been replaced by “arrest of filament elongation”.

40. Line 169: One of the most interesting observations is buried in the middle of a paragraph: “while 1 μ M twinfilin increased CP’s dissociation rate from BFC complex by \sim 11-fold, formin’s rate of dissociation did not change.”

Thank you for the kind words and suggestion. However, we feel moving it to the beginning or the end of the paragraph would be disruptive to data presentation.

41. Lines 187-200 belong in the discussion. Start the section with “To directly visualize the effects of twinfilin on the dynamics of BFC we used three-color single molecule TIRF imaging.”

Done.

42. Line 227: “About 73% of (the BFC) complexes (36 out of 49) transitioned to the 649-mDia1 state with 20 nM twinfilin...” Is 649-mDia1 state BF? If so, state this.

Yes, we have now edited the sentence as follows:

“About 73% of complexes (36 out of 49) transitioned to the 649-mDia1 BF state with 20 nM twinfilin as compared to 42% of complexes (30 out of 72) in control experiments”.

43. Lines 230-239: Belong in discussion.

Thank you for this suggestion. We have moved part of this text out of the results section.

44. Line 239: “destabilizes CP’s barbed end localization by binding side of actin filaments” needs a reference.

Thank you for pointing this out. We have now added the following reference: Wioland et al, 2017, Current biology.

45. Paragraph starting with line 245: The topic sentence is buried at the end “Even at high twinfilin concentration, we never observed any barbed ends with 649-mDia1 and 549-mTwf1 jointly bound to it.”

We have now deleted this entire paragraph and added the following text at the end of the next paragraph:

“Notably, in absence of CP, we never observed colocalization of 649-mDia1 and 549-mTwf1 at barbed ends. This explains our mf TIRF results that twinfilin doesn’t influence the rate of elongation or processivity of formins (Fig. 1h, Supplementary Fig. 1). Together, these observations indicate that the unlabeled CP molecule and twinfilin departed the barbed end simultaneously, leaving formin behind. Notably, we never observed twinfilin and CP simultaneously arrive at formin-occupied barbed ends.”

46. Lines 264-267: Two sentences say the same thing.

We have now combined the two sentences in to a single sentence as follows:

“Importantly, the departure of 549-mTwf1 from the decision complex led to an immediate transition of the barbed end from the arrested state to the fast formin-based elongation and translocation of the 649-mDia1 molecule.”

47. Lines 270-271: “implying that twinfilin’s primary role might be in accelerating dissociation of the BFC complexes rather than in preventing their formation.” This does not make sense. What is the evidence against preventing formation?

We have now removed this sentence.

48. Line 298: “CP on its own is capable of rapidly dissociating formin from [delete, formin-bound] barbed ends.” Should spell out what you mean by “rapid.”

We have reworded this sentence and replaced “rapid” by “increases” as follows:

“We find that CP on its own increases the rate of dissociation of formin from barbed ends”

49. Line 518 following: many problems with subscripts.

Thank you for pointing these out. They have now been fixed.

50. Line 768: Who is Ankita?

A member of the Homo sapiens species who also happens to be a lab member. She refuses to change her name.

REVIEWERS' COMMENTS

Reviewer #1 (Remarks to the Author):

This is a significantly improved version of a manuscript describing the combined effects of formin, capping protein and twinfilin on actin filament barbed end dynamics. The authors have satisfactorily addressed my previous concerns, and the manuscript is acceptable for publication after minor revisions.

1. Line 161: The authors should cite the Funk et al., (Nat. Comm., 2021) publication for the cryoEM structure of CP bound to the actin filament barbed end.
2. Lines 363-365: This sentence is a bit confusing, because there is no direct evidence demonstrating that in the BFCT complex, each factor would simultaneously interact with an actin filament barbed end (instead, it is more likely that formin, CP and twinfilin interact sequentially with the two terminal subunits of the actin filament barbed end). Thus, the authors should either delete or revise this sentence.
3. Fig 4: Labeling of panels 'd' and 'e' does not match with the figure legend. This should be corrected.

Reviewer #2 (Remarks to the Author):

This paper presents single filament/single molecule experiments to study how twinfilin and capping protein influence elongation of actin filament barbed ends associated with the formin mDia1. Previous work examined interactions of formin and CP as well as CP and twinfilin, but not the three proteins together. The authors are experts doing these experiments and present some valuable new data in Figs 3 and 4. The reviews of the original version of the paper raised concerns (reviewer 1) about overlap with prior research, the extent of the advances in this study and citations of previous work (I agree with their concerns) and problems with the presentation and interpretation (my review).

The authors provided detailed responses to each of my questions and concerns. Thanks to revisions of the text and figures and new experimental data, the paper is much improved. A substantially more concise presentation would make the paper much better.

I am satisfied with the authors' responses to all the numbered items in my review except the following:

1 and 19, Fig 6: Calling this a model or schematic is not the issue. The problem is that this detailed diagram gives the naïve reader the impression that we understand this complicated process. Therefore, it misrepresents the depth of the knowledge of mechanism, which will not be established without information about the reaction rates and the interactions of the proteins with each other and actin, along with computer simulations to test if the proposed reactions account for the experimental data. At this point, sticking with written descriptions of the main points is safer.

"CP was known to promote dissociation of formin mDia1 from the barbed end of an actin filament. Transient binding of twinfilin (~1s) to a barbed end occupied by mDia1 and CP, can dissociate CP, allowing elongation by the formin. While a single twinfilin binding event can displace CP from the trimeric complex, about 30 independent twinfilin binding events are required to remove capping protein alone from a barbed end. Thus, the actin filament depolymerase twinfilin can promote polymerization in the presence of both CP and formin."

A minor defect in this and other figures is that the subunits in the actin filaments are shown side by side rather than half staggered. Readers will conclude that the authors do not understand the structure of actin filaments.

3. Abstract: It is better after incorporating my suggestions, but the authors cannot resist using jargon such as "twinfilin acts as a pro-formin factor." "Pro-formin factor" does not have any accepted meaning.

4. Experimental conditions. The authors resisted, but one must include all experimental conditions in the figure legends. Readers should not have to calculate the concentrations themselves.

5, 6, 7: The authors do not understand the convention used in kinetics to describe rate constants (k 's) observed in experiments, which is to call these observed rate constants k_{obs} . The authors' corrections are wrong. These are "observed rate constants, k_{obs} " not "observed rates."

7. The replacement sentence is incorrect: it should read "Note that $k'-F$ and $k'-C$ are observed rate constants as they depend upon the concentration of twinfilin as well as the elongation rates of filaments associated with a formin." Furthermore, I do not understand " $k'-F$ " and " $k'-C$." I think that -F and -C should be subscripts.

17. It is hard to see how the authors can live with an “overestimation of (the) stock concentration of actin used in these particular experiments or a pipetting error.”

20. Is this problem with labeled actin explained in the text? Add on line 106 that using unlabeled actin avoids problems with labels on the actin.

21. Even if it worked, I hope the authors understand why pH 6.9 is not optimal for this reaction.

29 and lines 113-116, 130, 151, 154 and 157. Processivity. The authors use processive inappropriately to describe the lifetimes of formins on barbed ends. The authors write “(an) increase in (the rate of) dissociation of formin from the barbed end does lead to lower processivity.” Since processivity describes the continuous movements of a formin on an elongating actin filament or a motor on its track, I would say “(an) increase in (the rate of) dissociation of formin from the barbed end leads to shorter run lengths.” During these shorter runs, the formin is normally processive. The authors use an appropriate term, “barbed end residence time” of formins on actin filament ends to describe the experiments in Fig 1. I recommend using “residence time” rather than “processivity” throughout. For example, CP decreases the residence time but not the rates of the processive movements (Line 130).

32. You mean to say that you tested a range of concentrations, not that you varied the concentration during an individual experiment.

40. Just use this as the topic sentence to start a new paragraph on this interesting observation.

50. Usually one gives the last name of the helpful person.

New issues:

Line 14: “Formins accelerate elongation.” Elongation is faster with formins only in the presence of profilin; without profilin formins slow elongation. In any case they change the rate rather than “accelerating” elongation.

Lines 80-99: Describing all the results, figure by figure, in the introduction is not necessary. A short summary will suffice.

Lines 148-151 are not necessary.

Line 160: "X-ray diffraction and EM studies suggest that both CP and twinfilin interact with the last two subunits at the filament barbed end." This is slightly misleading. Ref 25 has an excellent structure of the complex of CP with two actins and one twinfilin. How twinfilin interacts with the barbed end of the filament is based on a comparison with CP on Arp1 in the dynactin complex, not a determined structure. The resolution is very low in ref 38. Therefore, the structure of the BFC is still far from established.

Line 167: Since "decision complex" is jargon, use "BFC" throughout.

Lines 198: The single molecule experiments in Figs 3 and 4 are first rate and novel, although the description in lines following line 260 could be much more direct. Just say "Experiments with 549-mTwf1 and 649-CP showed that Twf1 binding to a capped barbed end had a low probability of dissociating CP. Owing to photobleaching of 649-CP, we used the onset of depolymerization of barbed ends to detect when 549-mTwf1 dissociated unlabeled CP from ends. CP dissociation required on average 30.9 ± 6.5 successive 549-mTwf1 binding events, each lasting for 1.0 ± 0.1 s (Fig 4D)." Beautiful experiment.

Line 287: replace "Both twinfilin's CP-binding and actin-binding domains are necessary for its effects on BFC dynamics" with "Twinfilin must bind both CP and actin to dissociate the BFC."

Lines 305-315 repeat material from the introduction; delete. The rest of the discussion would be better if it were half as long and emphasized the new observations.

Tom Pollard

We thank the reviewers for their suggestions which have been extremely valuable in improving the manuscript. We are now submitting a revised version of the manuscript by taking into account reviewers' suggestions

Reviewer #1 (Remarks to the Author):

This is a significantly improved version of a manuscript describing the combined effects of formin, capping protein and twinfilin on actin filament barbed end dynamics. The authors have satisfactorily addressed my previous concerns, and the manuscript is acceptable for publication after minor revisions.

1. Line 161: The authors should cite the Funk et al., (Nat. Comm., 2021) publication for the cryoEM structure of CP bound to the actin filament barbed end.

We have now added additional citations for Funk et al (Nat. Comm., 2021) as well as the recently published Carman et al (Science, 2023) for cryoEM structure of CP bound to the actin filament barbed end.

2. Lines 363-365: This sentence is bit confusing, because there is no direct evidence demonstrating that in the BFCT complex, each factor would simultaneously interact with an actin filament barbed end (instead, it is more likely that formin, CP and twinfilin interact sequentially with the two terminal subunits of the actin filament barbed end). Thus, the authors should either delete or revise this sentence.

As we have already mentioned in our earlier response to the reviewer, we disagree with reviewer's opinion that there is no evidence for BFCT complex. The data presented here (and in two earlier studies: Shekhar et al, 2015 and Bombardier et al, 2015) actually shows that multiple proteins are simultaneously (and not sequentially) present at the barbed end. We would therefore like to leave the sentence unchanged.

3. Fig 4: Labeling of panels 'd' and 'e' does not match with the figure legend. This should be corrected.

Thank you for pointing this out. This has now been corrected.

Reviewer #2 (Remarks to the Author):

This paper presents single filament/single molecule experiments to study how twinfilin and capping protein influence elongation of actin filament barbed ends associated with the formin mDia1. Previous work examined interactions of formin and CP as well as CP and twinfilin, but not the three proteins together. The authors are experts doing these experiments and present some valuable new data in Figs 3 and 4. The reviews of the original version of the paper raised concerns (reviewer 1) about overlap with prior research, the extent of the advances in this study and citations of previous work (I agree with their concerns) and problems with the presentation and interpretation (my review).

The authors provided detailed responses to each of my questions and concerns. Thanks to revisions of the text and figures and new experimental data, the paper is much improved. A substantially more concise presentation would make the paper much better.

I am satisfied with the authors' responses to all the numbered items in my review except the following:

Thank you very much for your kind words and appreciation of the improvements in the manuscript. In line with the reviewer's suggestion, we have now further edited the text (especially in the introduction and discussion sections) to make it more concise.

1 and 19, Fig 6: Calling this a model or schematic is not the issue. The problem is that this detailed diagram gives the naïve reader the impression that we understand this complicated process. Therefore, it misrepresents the depth of the knowledge of mechanism, which will not be established without information about the reaction rates and the interactions of the proteins with each other and actin, along with computer simulations to test if the proposed reactions to account for the experimental data. At this point, sticking with written descriptions of the main points is safer.

"CP was known to promote dissociation of formin mDia1 from the barbed end of an actin filament. Transient binding of twinfilin (~1s) to a barbed end occupied by mDia1 and CP, can dissociate CP, allowing elongation by the formin. While a single twinfilin binding event can displace CP from the trimeric complex, about 30 independent twinfilin binding events are required to remove capping protein alone from a barbed end. Thus, the actin filament depolymerase twinfilin can promote polymerization in the presence of both CP and formin."

As we have previously pointed out, while we agree with the reviewer that there is room to make this figure more quantitative, the point of this figure in its current form is not to present an all-encompassing holistic picture of actin dynamics taking into account everything we know about these proteins. Instead, our aim is to present readers with a high-level schematic to help contextualize our findings. We would therefore like to keep the figure as it is.

A minor defect in this and other figures is that the subunits in the actin filaments are shown side by side rather than half staggered. Readers will conclude that the authors do not understand the structure of actin filaments.

Thank you for pointing this out, we apologize for this oversight. We have now corrected these issues in all of our figures.

3. Abstract: It is better after incorporating my suggestions, but the authors cannot resist using jargon such as "twinfilin acts as a pro-formin factor." "Pro-formin factor" does not have any accepted meaning.

"Pro-formin factor" in this context means a factor that promotes effects of formin. We note, this is exactly what twinfilin does in simultaneous presence of CP and formin, as stated in the abstract.

4. Experimental conditions. The authors resisted, but one must include all experimental conditions in the figure legends. Readers should not have to calculate the concentrations themselves.

The total concentrations of each protein used in every step of the experiment have been clearly stated in figure legends as well as in the methods section.

5, 6, 7: The authors do not understand the convention used in kinetics to describe rate constants (k 's) observed in experiments, which is to call these observed rate constants k_{obs} . The authors' corrections are wrong. These are "observed rate constants, k_{obs} " not "observed rates."

We have now made the necessary corrections in line with reviewer's advice.

7. The replacement sentence is incorrect: it should read "Note that $k'-F$ and $k'-C$ are observed rate constants as they depend upon the concentration of twinfilin as well as the elongation rates of filaments associated with a formin." Furthermore, I do not understand " $k'-F$ " and " $k'-C$." I think that $-F$ and $-C$ should be subscripts.

As advised by the reviewer, we have now added the following line in the text:

"Note that $k'-F + k'-C$ are observed rate constants as they depend upon the concentration of twinfilin as well as elongation rates of filaments associated with a formin."

17. It is hard to see how the authors can live with an "overestimation of (the) stock concentration of actin used in these particular experiments or a pipetting error."

We understand the reviewers' frustration. However, as stated earlier – the slightly faster elongation rates seen in figure 3C could have resulted from a number of experimental factors such as slight overestimation of actin stock concentration or activity. Importantly, the slightly faster elongation rate has no bearing on interactions between CP and formin at the barbed end of the filament. The results observed here are in complete agreement two previous studies, Shekhar et al, 2015 and Bombardier et al, 2015, which showed the formation of Formin-CP decision complex at the barbed end.

20. Is this problem with labeled actin explained in the text? Add on line 106 that using unlabeled actin avoids problems with labels on the actin.

We have now added the following line to the text:

"Using unlabeled actin prevents any artifacts that might arise from use of labeled actin."

21. Even if it worked, I hope the authors understand why pH 6.9 is not optimal for this reaction.

We understand.

29 and lines 113-116, 130, 151, 154 and 157. Processivity. The authors use processive inappropriately to describe the lifetimes of formins on barbed ends. The authors write "(an) increase in (the rate of) dissociation of formin from the barbed end does lead to lower processivity." Since processivity describes the continuous movements of a formin on an elongating actin filament or a motor on its track, I would say "(an) increase in (the rate of) dissociation of formin from the barbed end leads to shorter run lengths." During these shorter runs, the formin is normally processive. The authors use an appropriate term, "barbed end residence time" of formins on actin filament ends to describe the experiments in Fig 1. I recommend using "residence time" rather than "processivity" throughout. For example, CP decreases the residence time but not the rates of the processive movements (Line 130).

As previously pointed out, "processivity" is a commonly used term to describe formin's dwell times at the barbed end in a number of earlier studies including ours (please see Schmidt et al, 2021, PNAS; Shekhar et al, 2015, Nat. Comms.; Cao et al, 2018, eLife; Romero et al, 2007, JBC). We would therefore like to keep the text unchanged.

32. You mean to say that you tested a range of concentrations, not that you varied the concentration during an individual experiment.

Yes, this is exactly what we had intended. For clarity, we have now replaced all occurrences of “varying concentrations of” with “range of concentrations”.

40. Just use this as the topic sentence to start a new paragraph on this interesting observation.

Done.

50. Usually one gives the last name of the helpful person.

This person goes by one name only.

New issues:

Line 14: “Formins accelerate elongation.” Elongation is faster with formins only in the presence of profilin; without profilin formins slow elongation. In any case they change the rate rather than “accelerating” elongation.

While we agree with the reviewer that formins only increase the rate of elongation in presence of profilin, we do not think this extra level of detail is necessary for the abstract.

Lines 80-99: Describing all the results, figure by figure, in the introduction is not necessary. A short summary will suffice.

We believe this will help readers to get a quick overview picture of all of our key results.

Lines 148-151 are not necessary.

Done.

Line 160: “X-ray diffraction and EM studies suggest that both CP and twinfilin interact with the last two subunits at the filament barbed end.” This is slightly misleading. Ref 25 has an excellent structure of the complex of CP with two actins and one twinfilin. How twinfilin interacts with the barbed end of the filament is based on a comparison with CP on Arp1 in the dynactin complex, not a determined structure. The resolution is very low in ref 38. Therefore, the structure of the BFC is still far from established.

Thank you for pointing this out. We have now replaced this sentence with “X-ray diffraction and EM studies suggest that both CP and twinfilin interact with filament barbed ends”

Line 167: Since “decision complex” is jargon, use “BFC” throughout.

We have now replaced most instances of “decision complex” with “BFC”. However, we would like to retain the few remaining instances of the term “decision complex”. For the sake of clarity, we have also clearly defined in the text what the term “decision complex” refers to.

“CP and formin were initially thought to bind barbed ends in a mutually exclusive fashion. Contrary to this assumption, it was discovered that formin and CP simultaneously bind the same barbed end to form a so-called barbed end “decision complex”.”

Lines 198: The single molecule experiments in Figs 3 and 4 are first rate and novel, although the description in lines following line 260 could be much more direct. Just say “Experiments with 549-mTwf1 and 649-CP showed that Twf1 binding to a capped barbed end had a low probability of dissociating CP. Owing to photobleaching of 649-CP, we used the onset of depolymerization of barbed ends to detect when 549-mTwf1 dissociated unlabeled CP from ends. CP dissociation required on average 30.9 ± 6.5 successive 549-mTwf1 binding events, each lasting for 1.0 ± 0.1 s (Fig 4D).” Beautiful experiment.

Thank you for your kind words. We have now edited the text to make it more concise taking reviewer’s advice into consideration.

Line 287: replace “Both twinfilin’s CP-binding and actin-binding domains are necessary for its effects on BFC dynamics” with “Twinfiin must bind both CP and actin to dissociate the BFC.”

We understand the confusion. To reflect our results more accurately, we have now changed the title of the paragraph to “Twinfilin’s interaction with actin is essential for its effects on BFC dynamics”

Lines 305-315 repeat material from the introduction; delete. The rest of the discussion would be better if it were half as long and emphasized the new observations.

We have now removed repetitive text in the first paragraph to make it more concise.

Tom Pollard